# Temperature sensitive liposome based cancer nanomedicine enables tumour lymph node immune microenvironment remodelling

Shunli Fu[1], Lili Chang[1], Shujun Liu[1], Tong Gao[1], Xiao Sang[1], Zipeng Zhang[1], Weiwei Mu[1], Xiaoqing Liu[1], Shuang Liang[1], Han Yang[1], Huizhen Yang[1], Qingping Ma[1], Yongjun Liu [1] ✉ & Na Zhang [1] ✉

Targeting tumour immunosuppressive microenvironment is a crucial strategy in immunotherapy. However, the critical role of the tumour lymph node (LN) immune microenvironment (TLIME) in the tumour immune homoeostasis is often ignored. Here, we present a nanoinducer, NIL-IM-Lip, that remodels the suppressed TLIME via simultaneously mobilizing T and NK cells. The temperature-sensitive NIL-IM-Lip is firstly delivered to tumours, then directed to the LNs following pH-sensitive shedding of NGR motif and MMP2-responsive release of IL-15. IR780 and 1-MT induces immunogenic cell death and suppress regulatory T cells simultaneously during photo-thermal stimulation. We demonstrate that combining NIL-IM-Lip with anti-PD-1 significantly enhances the effectiveness of T and NK cells, leading to greatly suppressed tumour growth in both hot and cold tumour models, with complete response in some instances. Our work thus highlights the critical role of TLIME in immunotherapy and provides proof of principle to combine LN targeting with immune checkpoint blockade in cancer immunotherapy.

Immunotherapy has revolutionized antitumour therapy by harnessing the host immune system to identify and kill tumour cells[1,2]. The tumour microenvironment interacts with surrounding immune cells via the circulatory and lymphatic systems, which significantly affects the antitumour immune response and clinical efficacy[3,4]. Remodelling the immunosuppressive tumour microenvironment has been widely used to enhance immunotherapy[5–7]; however, the effects of immunotherapy still need improvement. Lymph node (LN) is the meeting grounds for dendritic cell (DC) antigen presentation and effective cytotoxic T lymphocyte (CTL) activation, showing strategic significance in tumour immunotherapy[8,9]. Nevertheless, the LNs microenvironment gradually becomes immunosuppressive under the stimulation of tumour

secretions as the tumour progresses, which greatly limits the antitumour immune response[10]. Although current LNs delivery strategies focus on vaccines[11,12], DC activation[13], and subcellular localization[14], insufficient attention has been given to the immunosuppressive LNs microenvironment in antitumour immunotherapy. Merely modulating the tumour immune microenvironment but ignoring the LNs immune microenvironment greatly limits the efficacy of antitumour immunotherapy[15,16]. Hence, we propose the concept of remodelling the tumour-lymph node immune microenvironment (TLIME) to simultaneously integrate the antitumour immune microenvironment, including the LNs immune microenvironment suitable for the activation of immune cells, with the tumour microenvironment suitable for

[1]NMPA Key Laboratory for Technology Research and Evaluation of Drug Products and Key Laboratory of Chemical Biology (Ministry of Education), Department of Pharmaceutics, School of Pharmaceutical Sciences, Cheeloo College of Medicine, Shandong University, Jinan, Shandong Province, China. ✉e-mail: liuyongjun@sdu.edu.cn; zhangnancy9@sdu.edu.cn

immune cells to exert killing effects on tumours. Eliminating suppressive factors and mobilizing positive immune cells in the TLIME may be a meaningful strategy to enhance antitumour immunotherapy.

Crucially, regulatory T cell (Treg) cause the immunosuppressive microenvironment in both tumours and LNs. Tregs disrupt DC antigen presentation, inhibit T- and B-cell proliferation and natural killer (NK) cell cytotoxicity, and secrete inhibitory cytokines (e.g., Transforming growth factor-β, TGF-β)[17,18]. Extensive efforts have been made to deplete Tregs and remodel the immunosuppressive microenvironment of tumour tissues, including the use of abemaciclib[19,20], an anti-CD25 mAb[21] or indoleamine 2,3-dioxygenase 1 (IDO1) inhibitors[22,23]. Moreover, immunosuppression in the LNs microenvironment has generally been attributed to the high expression of Tregs, and this high expression could limit the drainage of activated T cells from the LNs to the tumours[24,25]. We therefore hypothesized that inhibiting Treg cells in both tumours and LNs simultaneously may be a promising way to reshape the TLIME and improve the effects of immunotherapy.

After remodelling the TLIME, activating effective killer cells is another critical challenge. CTLs are the key immune cells in antitumour immunotherapy[26,27]. Immune checkpoint inhibitors, such as programmed cell death-1(PD-1) mAb, have been used to relieve the restrictions of CTLs[28], but the application of PD-1 mAb alone cannot maximize the tumour killing effects of CTLs[29]. Accumulating studies have shown that CTLs and NK cells could complement each other and that their coactivation could produce powerful antitumour effects, which is a promising means to enhance the efficacy of CTLs[30,31]. For example, the first tumour-targeting recombinant IL-15, Bj-001 injection, was initially indicated to be effective against advanced/metastatic solid tumours and was therefore further evaluated in clinical trials (NCT04294576). The premise was to achieve better therapeutic effects Bj-001 injection through sufficient infiltration of CTLs and NK cells. Current studies have focused on the immunogenic cell death (ICD) effect and chemokines to promote the intratumoural infiltration of CTLs and NK cells[32,33]. Moreover, targeted delivery of vaccines to LNs could more effectively promote intratumoural infiltration[34]. For example, we have proven that the targeted delivery of micelles coloaded with vaccine and IL-15 significantly promoted the infiltration and activation of CTLs and NK cells in tumours[35]. Therefore, simultaneously coactivating CTLs and NK cells in the TLIME might be a meaningful path that is worth exploring to obtain an ideal antitumour effect.

Thus, we developed a strategy to remodel the suppressed TLIME and simultaneously mobilize T and NK cells in the TLIME to achieve optimal efficacy (Fig. 1). A lipid-based nanoinducer by co-loading photothermal agent IR780 and IDO1 inhibitor 1-MT into the temperature-sensitive liposome with the modification of pH-sensitive NGR and matrix metalloproteinase 2 (MMP2) responsive interleukin (IL)-15 (NGR/IL-15-IR780/1-MT-Lip, NIL-IM-Lip) was meticulously constructed to precisely target tumour tissues and be directed to the LNs as expected. Then, we optimized the particle size and cholesterol ratio of NIL-IM-Lip for LNs accumulation[36,37]. Effective TLIME remodelling required the efficient tumour and LNs accumulation of agents. To guarantee the tumour accumulation and LNs directing properties of NIL-IM-Lip, the pH-sensitive target DSPE-Acylhydrazone bond-PEG$_{5000}$-NGR (DSPE-HyD-PEG$_{5000}$-NGR) was used to provide tumour-specific targeting and quick disassembly in the tumour microenvironment to meet the needs for LNs allocation. Then, to coactivate CTLs and NK cells, the photothermal agent IR780 was selected to trigger photothermal therapy to induce the ICD effect and the IL-15 was used to promote the killing ability of immune cells[38]. Considering the different target cells of IR780 and IL-15, the MMP2-sensitive linker DSPE-PEG$_{2000}$-MMP2 sensitive peptide-Maleimide (DSPE-PEG$_{2000}$-MMP2 sensitive pep-Mal) was used as linker to modify IL-15 on the surface of NIL-IM-Lip, which could achieve the xenotype cell delivery and release IL-15 in the tumour microenvironment with enzyme MMP2

to strengthen the coactivation of CTLs and NK cells. Moreover, the IDO1 inhibitor 1-MT was selected to relieve Treg inhibition in the TLIME and reverse the immunosuppression TLIME. 1-MT also simultaneously eliminated the overexpression of IDO1 induced by photothermal therapy[39]. Finally, in order to enhance the antitumour efficacy and reduce the toxicity and side effects on normal tissues, the light-controlled release and spatiotemporal specific release was designed and the temperature-sensitive lipid material 1,2-dipalmitoyl-sn-glycero-3-phosphocholine (DPPC) used as a carrier component to achieve the thorough and specific drug release[40].

Here, the nanomodulatory inducer NIL-IM-Lip is designed to target both tumours and LNs for remodelling the TLIME and enhancing the CTL and NK cells antitumour effects. After combination with anti-PD-1 therapy, the NIL-IM-Lip effectively inhibits the tumour growth in both hot and cold tumours (B16F10 and CT26 model). The nanomodulatory inducer offers a platform that can provide delivery to both tumours and LNs and enables robust antitumour immunotherapy against both hot tumours and cold tumours while remodelling the suppressive TLIME, thereby paving the way for further applications.

## Results

### NIL-IM-Lip with a small size and 1/8 mass ratio of cholesterol exhibited improved tumour accumulation and LNs directing abilities

In this study, pH-sensitive DSPE-Hyd-PEG$_{5000}$-NGR (77.72%) and the DSPE-PEG$_{2000}$-MMP2-sensitive pep-Mal (82.8%) were successfully synthesized and characterized using $^1$H nuclear magnetic resonance ($^1$H-NMR) (Supplementary Figs. 1-3). We prepared a series of pH/MMP2/temperature triple-sensitive nanoinducers with different particle sizes and mass ratios of cholesterol (Fig. 2a). Briefly, we prepared the DPPC liposome without drug loading and modification of DSPE-Hyd-PEG$_{5000}$-NGR and DSPE-PEG$_{2000}$-MMP2-sensitive pep-IL-15 (Blank-Lip), the DPPC liposome co-loading IR780 and 1-MT (IR780/1-MT-Lip, IM-Lip) and the DPPC liposome co-loading IR780 and 1-MT and modification of DSPE-Hyd-PEG$_{5000}$-NGR and DSPE-PEG$_{2000}$-MMP2-sensitive pep-IL-15 (NGR/IL-15-IR780/1-MT-Lip, NIL-IM-Lip). The large size of Blank-Lip, IM-Lip, NIL-IM-Lip was described as Blank-Lip-L, IM-Lip-L, NIL-IM-Lip-L. The small size of Blank-Lip, IM-Lip, NIL-IM-Lip was described as Blank-Lip-S, IM-Lip-S, NIL-IM-Lip-S. The Blank-Lip-S and Blank-Lip-L were approximately 42.60 nm and 115.0 nm, respectively. After loading IR780 and 1-MT, the sizes of IM-Lip-S and IM-Lip-L increased to 47.54 nm and 122.4 nm, respectively. Then, the fabricated NIL-IM-Lip-S and NIL-IM-Lip-L displayed further increases in size to 65.75 nm and 139.1 nm after connection with the DSPE-Hyd-PEG$_{5000}$-NGR and DSPE-PEG$_{2000}$-MMP2-sensitive pep-IL-15, respectively (Fig. 2b). We used near infrared fluorescence (NIRF) imaging to evaluate the tumour accumulation and LNs directing properties of NIL-IM-Lip. Considering of the IR780 could be used for NIRF imaging, we prepared the large size (NGR-IR780-Lip-Large, N-I-Lip-L) and small size (NGR-IR780-Lip-Small, N-I-Lip-S) DPPC liposome loading IR780 and modification of DSPE-Hyd-PEG$_{5000}$-NGR. We determined the influence of particle size on LNs accumulation (Supplementary Table 1). Specifically, the fluorescence signals of N-I-Lip-L and N-I-Lip-S in LNs gradually increased with time. N-I-Lip-S accumulated in LNs accounting for 0.91% (12 h), 3.69% (24 h), and 6.83% (48 h), while the amount of N-I-Lip-L located in LNs decreased to 0.25% (12 h), 1.25% (24 h), and 3.37% (48 h), respectively. The fluorescence signal of the tumour tissues in both N-I-Lip-L and N-I-Lip-S showed no significant difference (Fig. 2c, Supplementary Fig. 4). N-I-Lip-S showed stronger LNs directing properties than N-I-Lip-L after intravenous injection, which was consistent with the belief that small nanoparticles accumulate better in the LNs after intravenous injection[41] (Fig. 2d, Supplementary Fig. 4). Therefore, N-I-Lip-S was chosen for use in this study. We next studied the effects of cholesterol transport on LNs accumulation capacity. The NIRF image and fluorescence quantification results showed the better

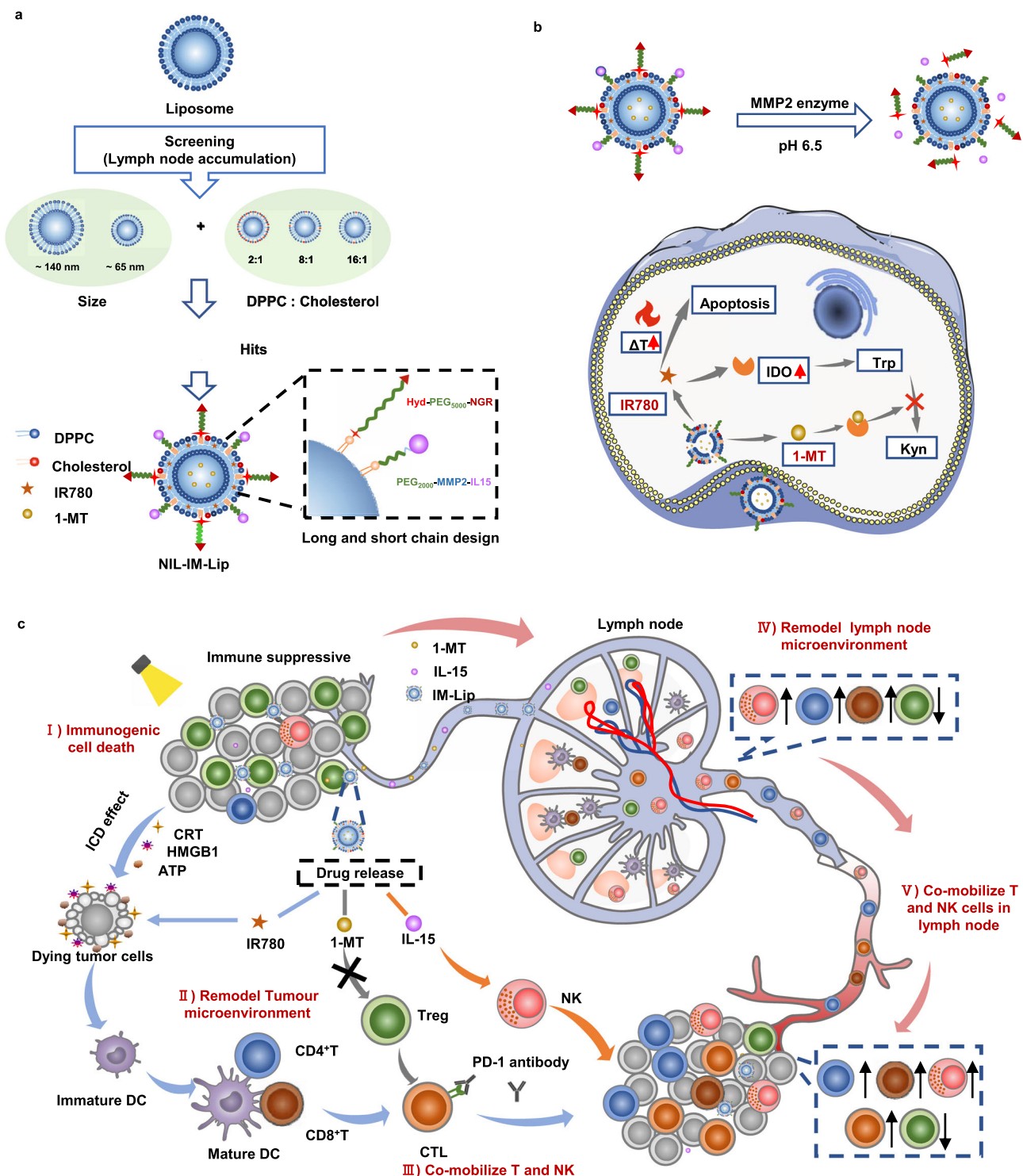

**Fig. 1 | Schematic illustration of nanoinducers that remodel the immunosuppressive tumour-lymph node microenvironment to mobilize T and NK cells.**
**a** Screening the size and cholesterol mass ratio for LNs accumulation. The pH/MMP2/temperature-sensitive nanoinducers (NGR/IL-15-IR780/1-MT-Lip, NIL-IM-Lip) were prepared with DPPC (temperature sensitive), IR780, 1-MT, DSPE-PEG$_{2000}$-MMP2-sensitive pep-IL-15 (MMP2 sensitive) and DSPE-Hyd-PEG$_{5000}$-NGR (pH sensitive). **b** After accumulation in tumour tissues, the nanoinducers release IL–15 and NGR, which are then taken up by tumour cells. IR780 induces tumour cell apoptosis via hyperpyrexia and enhances the higher expression of IDO1. 1-MT inhibits kynurenine accumulation under photothermal therapy. **c** Nanoinducers can be targeted to tumour tissues and directed to LNs. (i) Photothermal therapy reduces tumour burden and elicits ICD to enhance the immune response. (ii) The NIL-IM-Lip accumulated in the tumours comobilized CTLs and NK cells by combining the ICD effect, IL-15 and a PD-1 mAb. (iii) 1-MT was further used to inhibit Treg infiltration in tumours to remodel the tumour microenvironment. (iv) Nanoinducers directed to the LNs comobilized CTLs and NK cells. (v) Nanoinducers directed to the LNs remodelled the LNs microenvironment via the inhibition of Tregs. HyD-PEG$_{5000}$-NGR: Acylhydrazone bond-PEG$_{5000}$-NGR; PEG$_{2000}$-MMP2-IL-15: PEG$_{2000}$-MMP2 sensitive peptide-IL-15; IM-Lip IR780/1-MT-Liposome, Trp tryptophan, Kyn kynurenine.

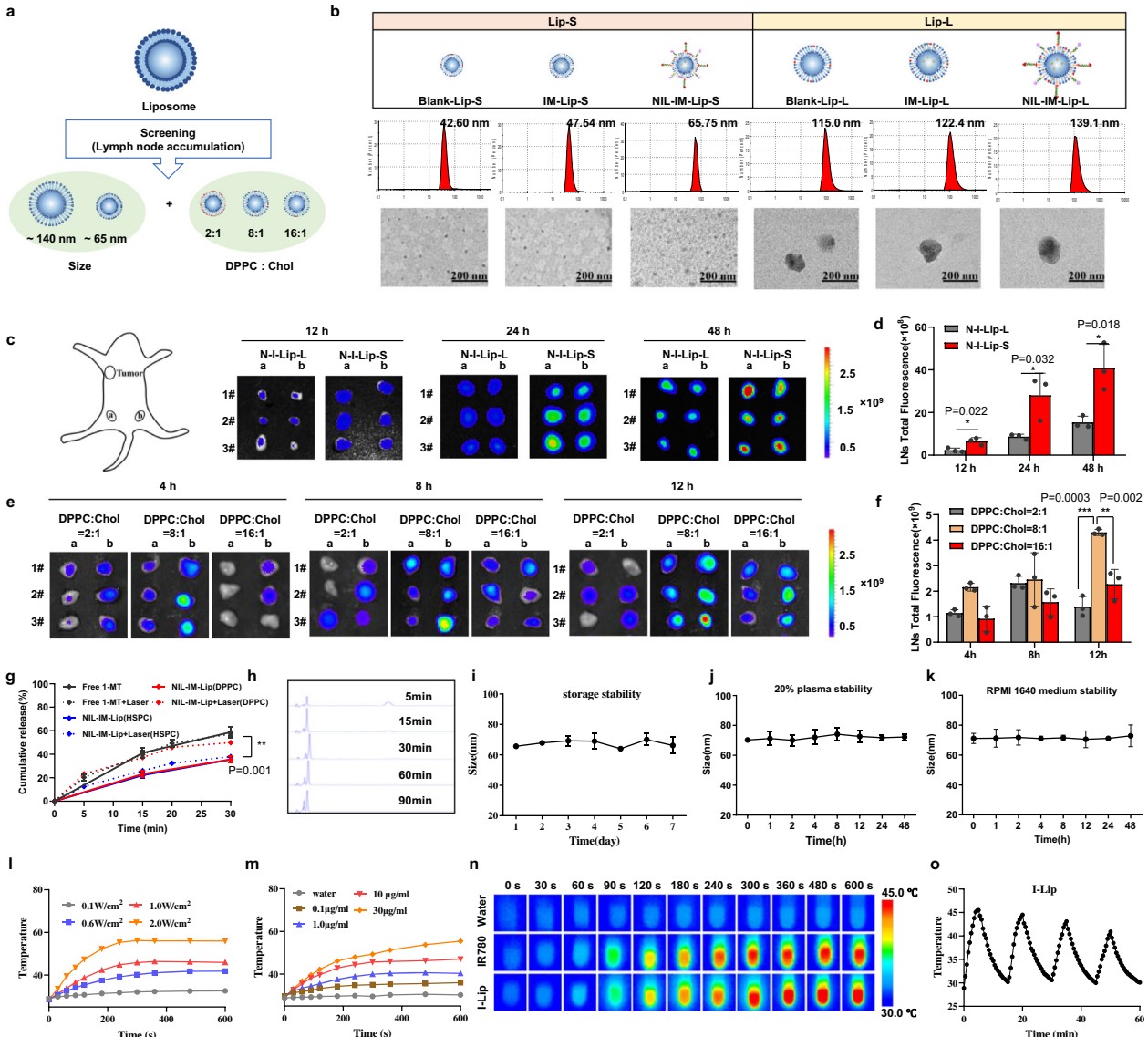

**Fig. 2 | Efficacy of NIL-IM-Lip with a small size and 1/8 mass ratio of cholesterol on LNs directing properties, xenotype cell delivery and photothermal conversion. a** Schematic illustration of the particle size and cholesterol mass ratio screenings for LNs accumulation. **b** Particle size and morphology of different formulations (n = biologically independent experiments). **c** Ex vivo imaging of LNs at 12, 24, 48 h post-administration N-I-Lip-L and N-I-Lip-S (1#, 2#, 3# represent three mice, n = 3 biologically independent experiments). **d** Total fluorescence intensities of the LNs at 12, 24, 48 h post-administration of N-I-Lip-L and N-I-Lip-S (n = 3 biologically independent experiments). **e, f** Ex vivo imaging **e** and total fluorescence intensity **f** of the LNs at 4, 8, and 12 h post-administration of N-I-Lip with different cholesterol mass ratios (n = 3 biologically independent experiments). **g** Cumulative release curves of 1-MT from different groups treated with and without laser irradiation (808 nm) for 5 min in PBS (pH 7.4) (n = 3 biologically independent experiments). **h** HPLC chromatograms of 1.0 mg/mL MMP2-sensitive peptide after incubation with 50 μg/mL MMP2 enzyme for different lengths of time. **i–k** NIL-IM-

Lip showed good storage stability **i**, 20% plasma stability **j** and RPMI 1640 medium stability (**k**) (n = 3 biologically independent experiments). **l** Temperature profile of I-Lip (10 μg/mL) with laser irradiation (808 nm) at different powers (n = 3 biologically independent experiments). **m** Temperature profiles of different concentrations of I-Lip after irradiation with a laser irradiation (808 nm) at 1.0 W/cm² (n = 3 biologically independent experiments). **n** Infrared thermal imaging of water, IR780 and I-Lip at different time points after irradiation with a laser irradiation (808 nm) at 1.0 W/cm² for 10 min. **o** The temperature profile of I-Lip recorded after four laser on/off cycles (808 nm). Data are presented as mean values ± SD. Statistical significance was calculated by the two-tailed Student's t-test **d**, **g** and one-way ANOVA analysis of variance with Tukey's post hoc test **f**. *P < 0.05, **P < 0.01, ***P < 0.001. Source data are provided as a Source Data file. Blank-Lip-S/L Blank-Lip-Small/Large, IM-Lip-S/L IR780/1-MT-Lip-Small/Large, NIL-IM-Lip-S/L NGR/IL-15-IR780/1-MT-Lip-Small/Large, N-I-Lip-L/S NGR-IR780-Lip-Small/Large, DPPC:Chol: the mass ratio of DPPC and cholesterol.

fluorescent signal in the LNs and higher ratio of LNs/Total fluorescence under 1/8 mass ratio of cholesterol (Supplementary Table 1, Supplementary Fig. 5, Fig. 2e, f). Notably, we found a clear transport pathway between the tumour and LNs after injection of N-I-Lip (Supplementary Fig. 6), suggesting that our hypothesis of considering the TLIME may be meaningful in antitumour immunotherapy. Hence, we selected the NIL-IM-Lip with small size and a 1/8 mass ratio of cholesterol and described as NIL-IM-Lip for subsequent experiments, which targeted

tumours and was directed to the LNs, as expected, to obtain ideal immune cell regulation. Interestingly, we also evaluated the effects of cholesterol transport on LNs accumulation capacity in other lipid material (lecithin high potency and egg yolk lecithin) (Supplementary Fig. 7). The lecithin high potency and egg yolk lecithin liposomes showed the similar results as DPPC liposome, indicating the great LNs accumulation of liposome with a 1/8 mass ratio of cholesterol (DPPC, lecithin high potency and egg yolk lecithin).

Considering that previous studies have demonstrated that photothermal therapy treatment could induce the overexpression of IDO1, we evaluated the expression of IDO1 in tumours and LNs. Compared with the NS group and IR780 + Laser (L) (IR780 + L) group, the N-I-Lip +Laser (N-I-Lip+L) group upregulated the expression of IDO1 in both tumour tissues and LNs (Supplementary Fig. 8). These results confirmed our hypothesis that inhibiting Treg infiltration in the TLIME by using an IDO1 inhibitor (1-MT) was necessary.

Moreover, we characterized the properties of the selected NIL-IM-Lip, which were listed in Supplementary Table 2. The encapsulation efficiencies of NIL-IM-Lip were $81.8 \pm 2.8\%$ (IR780) and $30.5 \pm 3.2\%$ (1-MT). For further verifying the successful preparation of NIL-IM-Lip, the maximum absorption wavelength of IR780 shifted from 783 nm to 806 nm after its encapsulation into NIL-IM-Lip, which may be related to the change in environment, from polar to a nonpolar hybrid matrix. The maximum absorption wavelength of free 1-MT in water was 287 nm, and this value did not change significantly after 1-MT was embedded in NIL-IM-Lip because of the consistent residence environment (Supplementary Fig. 9). Subsequently, we investigated the release behaviour of 1-MT. The cumulative release rates of 1-MT in the free 1-MT group and free 1-MT with laser group were 58.82% and 57.92%, respectively, indicating that the laser had no effect on the release behaviour of free 1-MT. The cumulative 1-MT release in the NIL-IM-Lip group (35.51%) was significantly lower than that in the NIL-IM-Lip with laser group (49.90%) ($P < 0.01$) (Fig. 2g), which may attribute the disruption of liposome after the irradiation. For further verify whether the liposome was disrupted, the morphology of NIL-IM-Lip after irradiation of laser was evaluated and the results in Supplementary Fig. 10 show the liposomes were destroyed after the irradiation of laser. Moreover, the temperature non-sensitive lipid Hydrogenated Soybean Phospholipid (HSPC) was used to demonstrate the temperature sensitive release of 1-MT under the irradiation of laser. The 1-MT release results (Fig. 2g) of NIL-IM-Lip (HSPC) group and NIL-IM-Lip+Laser (HSPC) group showed the similar cumulative release, which may because the temperature could not reach the transformation temperature of HSPC after the irradiation of laser. All these results showed that more 1-MT was fastly released from NIL-IM-Lip (DPPC) with laser irradiation than that in the NIL-IM-Lip (DPPC) without laser irradiation, which may be contribute to the disrupted of NIL-IM-Lip (DPPC) after the irradiation of laser.

## NIL-IM-Lip enhanced xenotype cell delivery and photothermal conversion efficacy

The DSPE-PEG$_{2000}$-MMP2 sensitive pep-Mal was used as a linker to achieve xenotype cell delivery efficacy. To verify the cleavage of the MMP2-sensitive peptide, we selected a series of enzyme concentrations (0, 0.5, 1, 5, 10, 50 μg/mL) for coincubation with the MMP2-sensitive peptide. The results showed that the MMP2-sensitive peptide could be completely degraded after incubation with 50 μg/mL MMP2 enzyme for 12 h (Supplementary Fig. 11). So, 50 μg/mL MMP2 enzyme was used in the subsequent experiment. Then, the incubation time with the MMP2 enzyme was further examined. The results of Fig. 2h show that the MMP2-sensitive peptide could be completely degraded in 0.5 h with 50 μg/mL MMP2 enzyme. Therefore, the follow-up experiments were performed with 50 μg/mL MMP2 enzyme for 0.5 h. Moreover, the particle size of NIL-IM-Lip changed minimally, indicating that NIL-IM-Lip possessed good stability in storage, 20% plasma and RPMI 1640 medium (Fig. 2i–k).

Next, we investigated the photothermal conversion performance of IR780-Lip (I-Lip) under irradiation with an 808 nm laser. The power of the laser and the concentration of IR780 greatly influenced the photothermal conversion efficacy. I-Lip (10 μg/mL IR780) increased from 29 °C to 32.6, 41.8, 45.9 and 56.0 °C at powers of 0.1, 0.6, 1.0 and 2.0 W/cm$^2$ (10 min), respectively (Fig. 2l). After

that, the photothermal conversion efficacies with different concentrations of I-Lip were further determined. The temperature increased from approximately 29 °C to 30.2, 40.3, 47.0, and 55.5 °C in 10 min at I-Lip concentrations of 0, 1, 10 and 30 μg/mL, respectively (Fig. 2m). Considering the safety and efficacy of photothermal therapy, 1.0 W/cm$^2$ and 10 μg/mL IR780 were used for the follow-up experiments. The infrared thermal imaging results (Fig. 2n) were consistent with the above results. All of these results indicated that the photothermal conversion effect was positively correlated with the concentration of IR780 and the power of the 808 nm laser. Furthermore, after four laser on/off cycles, no significant change in photothermal conversion efficacy was observed in either the free IR780 group or I-Lip group, indicating good photothermal stability (Fig. 2o and Supplementary Fig. 12).

## NIL-IM-Lip displayed great tumour and LNs accumulation and local hyperthermia efficacy

B16F10 tumour-bearing C57BL/6 mice were used to evaluate the accumulation of NIL-IM-Lip in the tumours and LNs. The imaging (Fig. 3a, b) and quantitative fluorescence signal data (Fig. 3c) showed that the N-I-Lip group had stronger tumour accumulation than the IR780 group and I-Lip group. The N-I-Lip group exhibited more LNs accumulation than the IR780 group. Considering the effect of the modification of IL-15 on the surface of liposome, we also compared the biodistribution of N-I-Lip group and NGR/IL-15-IR780-Lip (NIL-I-Lip). As shown in Supplementary Fig. 13, the N-I-Lip and NIL-I-Lip showed the similar tumour and LNs accumulation. Hence, the N-I-Lip was used for the subsequent experiment to replace the NIL-I-Lip. These results indicated that NIL-IM-Lip possessed greater tumour accumulation and LNs directing abilities than the free drug.

After accumulating in the tumour of NIL-IM-Lip, the in vivo photothermal conversion performance of NIL-IM-Lip was further evaluated. After 12 h, N-I-Lip effectively accumulated in the tumour (Fig. 3a), at which point mice were irradiated with an 808 nm laser and thermal images were captured. The temperatures of the local tumour tissues in the NS group, IR780 group, I-Lip group and N-I-Lip group increased successively after 10 min of irradiation, which may positively correlate with the tumour site accumulation capacities of the different formulations. The temperatures of the local tumour tissues in the N-I-Lip group increased to 38.47 °C, 44.03 °C, and 52.83 °C at laser powers of 0.6, 1.0, and 2.0 W/cm$^2$, respectively (Fig. 3d, e, Supplementary Fig. 14). Considering that effective tumour cell necrosis induction occurs at approximately 42−45 °C, 1.0 W/cm$^2$ was chosen as the laser power for the in vivo studies.

Then, we also evaluated the distribution of IL-15. Cy5.5-BSA was selected to simulate IL-15 for NIRF imaging. At 12 h, NIL-IM-Lip modified with MMP2-sensitive Cy5.5-BSA (MMP2-sensitive) showed stronger LNs accumulation than NIL-IM-Lip modified with MMP2-non-sensitive Cy5.5-BSA (MMP2-non-sensitive) and Cy5.5-BSA (Fig. 3f, g), indicating that the functions of IL-15 were not affected by the laser because of its good accumulation in the LNs before irradiation.

Above all, the free drug did not show significant specific tumour and LNs accumulation, however, the NIL-IM-Lip exhibited good tumour accumulation and LNs directing abilities, which could better regulate immune cell function in the TLIME.

## NIL-IM-Lip+L promoted tumour cell apoptosis, pH-sensitive disassembly and deep tumour penetration for a stronger antitumour effect in vitro

The apoptosis rate of NIL-IM-Lip was studied in B16F10 cells. As shown in Fig. 4a, the NIL-IM-Lip+Laser (NIL-IM-Lip+L) group showed higher apoptosis rates than the IR780 + L, IR780 + 1-MT + Laser (IR780 + 1-MT + L) and IR780 + 1-MT + IL-15+Laser (IR780 + 1-MT + IL-15 + L). Then, the antitumour efficacy in vitro was further evaluated by methyl thiazolyl tetrazolium (MTT) assay. Irradiation with laser

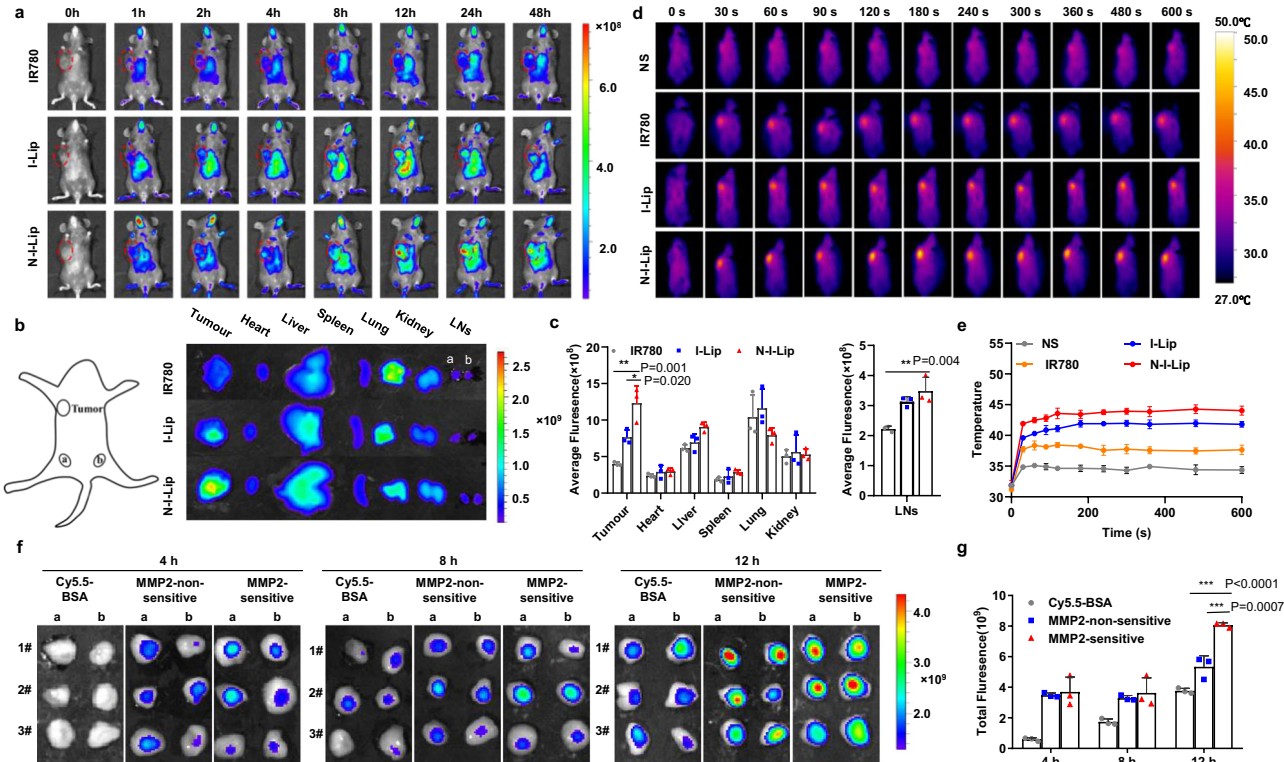

**Fig. 3 | Accumulation of NIL-IM-Lip in the tumour and LNs and the local hyperthermia effects. a** In vivo biodistribution imaging of the mice after injection of different formulations (i.v.). **b** Ex vivo imaging of the tumour, main organs and LNs at 48 h. **c** Average fluorescence intensities of the tumour, main organs and LNs at 48 h ($n = 3$ biologically independent experiments). **d** Infrared thermal images of mice treated with NS, IR780, I-Lip and N-I-Lip under $1.0 \, \text{W/cm}^2$ laser irradiation (808 nm). **e** Temperature profiles of the mouse tumours after treatment with NS, IR780, I-Lip and N-I-Lip under $1.0 \, \text{W/cm}^2$ laser irradiation (808 nm) ($n = 3$ biologically independent experiments). **f** Ex vivo imaging of LNs at 4, 8, and 12 h post-administration of Cy5.5-BSA, MMP2-non-sensitive and MMP2-sensitive. Cy.5.5-BSA was used to replace IL-15 for NIRF imaging ($n = 3$ biologically independent experiments). **g** Total fluorescence intensities of the LNs at 4, 8, and 12 h post-administration of Cy5.5-BSA, MMP2-non-sensitive and MMP2-sensitive ($n = 3$ biologically independent experiments). The data are shown as the mean ± SD ($n = 3$ biologically independent experiments). Statistical significance was calculated by the one-way ANOVA analysis of variance with Tukey's post hoc test. *$P < 0.05$, **$P < 0.01$, ***$P < 0.001$. Source data are provided as a Source Data file. I-Lip: IR780-Lip; N-I-Lip: NGR-IR780-Lip; Cy5.5-BSA: Cy5.5 labelled BSA; MMP2-non-sensitive: IL-15 modified on the NIL-IM-Lip by MMP2-non-sensitive peptide; MMP2 sensitive: IL-15 modified on the NIL-IM-Lip by MMP2 sensitive peptide.

alone induce no obvious cytotoxicity, revealing the critical role of IR780 (Fig. 4b). NIL-IM-Lip+L treatment showed higher cytotoxicity than IR780 + L treatment. Moreover, cell viability in the Blank-Lip, 1-MT, IL-15, IR780 and NIL-IM-Lip groups was greater than 80%, suggesting the safety of these formulations without laser irradiation (Fig. 4c).

In our study, NGR was connected to the surface of NIL-IM-Lip by pH-sensitive Hyd bonds, which could quickly disassemble from the surface of NIL-IM-Lip in the tumour microenvironment. The FITC was used to replace 1-MT to prepare FITC-Lip (F-Lip) and NGR-FITC-Lip (N-F-Lip). Treatment with N-F-Lip gave a higher uptake ratio than F-Lip and N-F-Lip treatment after preincubation at pH 6.5 (Fig. 4d, e, $P < 0.001$), suggesting the stronger targeting efficacy of N-F-Lip. There was no difference in the fluorescence of F-Lip and N-F-Lip after preincubation at pH 6.5, confirming that DSPE-Hyd-PEG$_{5000}$-NGR had pH-sensitive disassembly properties. In addition, the images of HUVECs (Fig. 4f) showed the fluorescence of each group increased as the incubation time increased, showing a time-dependent change. These results verified that NIL-IM-Lip presented enhanced tumour accumulation and pH-sensitive disassembly properties.

Then, we used 3D tumour spheroids to verify deep in tumour tissue penetration. The FITC-Lip-Small particle size (F-Lip-S) group showed stronger fluorescence in the tumour interior than the FITC-Lip-Large particle size (F-Lip-L) group (Fig. 4g), which was attributed to the small particle size of F-Lip-S.

## NIL-IM-Lip+L promoted DC maturation by inducing the ICD effect in vitro

Photothermal therapy can induce ICD effects by promoting the exposure of calreticulin (CRT), the secretion of high mobility group box 1(HMGB1) and the production of adenosine triphosphate (ATP) and reactive oxygen species (ROS). Hence, the levels of CRT, HMGB1, ATP and ROS were selected as indices to evaluate the ICD effect. NIL-IM-Lip+L treatment generated higher levels of CRT exposure and fewer HMGB1 nuclei than IR780 + 1-MT + L and IR780 + 1-MT + IL-15 + L treatment ($P < 0.001$) (Fig. 5a–d). Quantitative analysis of HMGB1 in the cell culture supernatant revealed that the NIL-IM-Lip+L group had higher HMGB1 secretion than the IR780 + 1-MT + L group and IR780 + 1-MT + IL-15 + L group ($P < 0.01$) (Fig. 5e). The ATP assay of the cell culture supernatant gave the same results (Fig. 5f). Furthermore, ROS generation was also detected. The groups with irradiation of laser showed significantly increased fluorescence compared with the groups without laser irradiation. The NIL-IM-Lip+L group showed higher fluorescence than the IR780 + 1-MT + L group and IR780 + 1-MT + IL-15 + L group (Supplementary Fig. 15). All of these results illustrated that NIL-IM-Lip+L treatment could promote stronger CRT exposure, HMGB1 secretion, and ATP and ROS generation to promote more DC maturation.

The signal released by the ICD effect could promote the expression of CD80 and CD86. DCs are one of the most important antigen-presenting cells that thereby promote the activation of T cells. DC maturation in vitro ability was evaluated with a transwell system

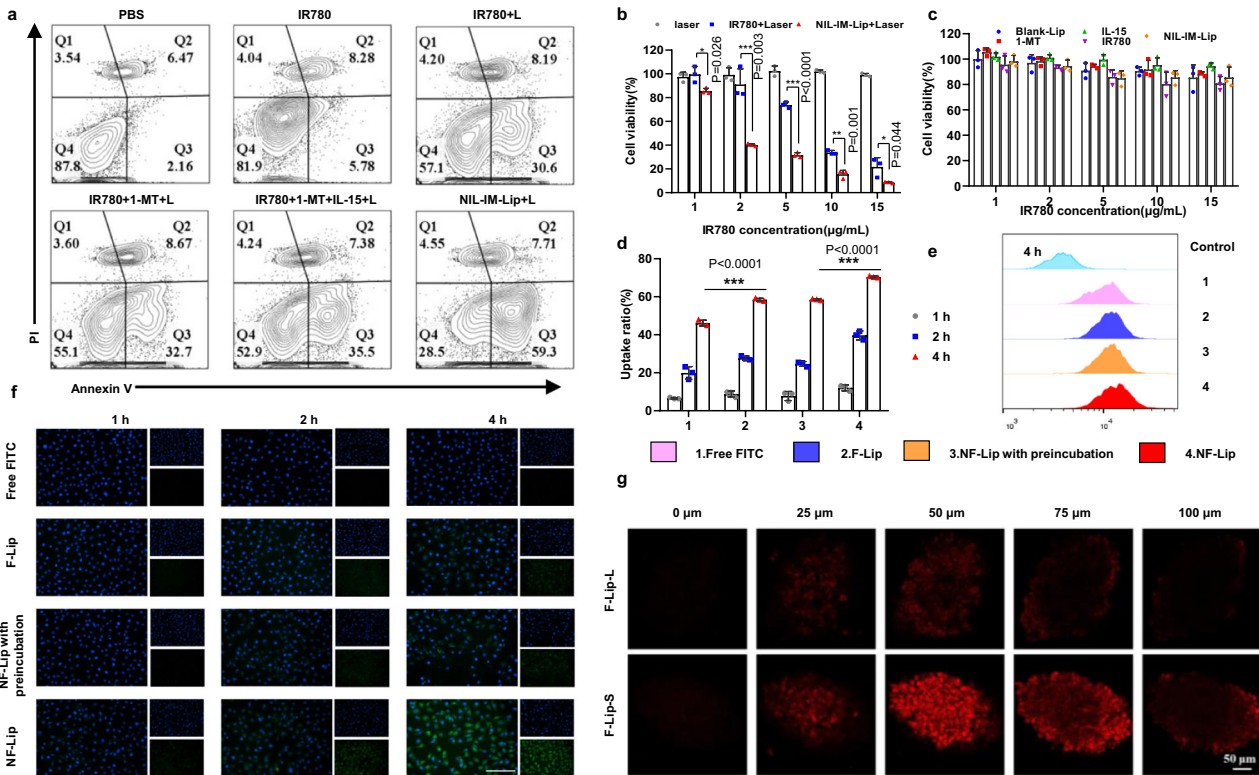

**Fig. 4 | The in vitro antitumour effects of NIL-IM-Lip on apoptosis, pH-sensitive disassembly and deep tumour penetration. a** Flow cytometric apoptosis analysis of B16F10 cells after treatment with PBS, IR780, IR780 + L, IR780 + 1-MT + L, IR780 + 1-MT + IL-15 + L and NIL-IM-Lip+L. The IR780 + L, IR780 + 1-MT + L, IR780 + 1-MT + IL-15 + L and NIL-IM-Lip+L groups were irradiated with an 808 nm laser for 5 min (1.0 W/cm²). **b** Viabilities of B16F10 cells after incubation and treatment with Laser, IR780 + Laser and NIL-IM-Lip+Laser. The IR780 + Laser and NIL-IM-Lip+Laser groups were irradiated with an 808 nm laser for 5 min (1.0 W/cm²) (*n* = 3 biologically independent experiments). **c** Viabilities of B16F10 cells after incubation with Blank-Lip, IL-15, 1-MT, IR780 and NIL-IM-Lip (*n* = 3 biologically independent experiments). **d**, **e** Flow cytometric analysis of HUVEC uptake (*n* = 3 biologically independent experiments). FITC was selected to replace 1-MT for the

cellular uptake assay. **f** Fluorescence microscopy image of HUVEC uptake (*n* = 3 biologically independent experiments). Scale bar = 200 μm. **g** F-Lip-S showed stronger deep tumour penetration than F-Lip-L in B16F10 tumour spheres (*n* = 3 biologically independent experiments). Scale bar = 50 μm. Data are presented as mean values ± SD. Statistical significance was calculated by the one-way ANOVA analysis of variance with Tukey's post hoc test. *\*P* < 0.05, *\*\*P* < 0.01, *\*\*\*P* < 0.001. Source data are provided as a Source Data file. IR780 + L IR780 + Laser, IR780 + 1-MT + L IR780 + 1-MT + Laser, IR780 + 1-MT + IL-15 + L IR780 + 1-MT + IL-15+Laser, NIL-IM-Lip+L NGR/IL-15-IR780/1-MT-Lip+Laser, F-Lip FITC-Lip, F-Lip with pre-incubation: FITC-Lip with preincubation under pH 6.5; NF-Lip NGR-FITC-Lip, F-Lip-L/S FITC-Lip-Large/Small.

(Fig. 5g). Mature DCs were marked as CD11c⁺CD80⁺CD86⁺ cells. The NIL-IM-Lip+L group displayed the greatest maturation of DCs (Fig. 5h, i). Moreover, the NIL-IM-Lip+L group secreted a higher level of Tumour necrosis factor (TNF)-α into the culture supernatant than the IR780 + 1-MT + L group and IR780 + 1-MT + IL-15 + L group (*P* < 0.05) (Fig. 5j). Taken together, these results showed that NIL-IM-Lip+L could greatly induce the ICD effect and promote the maturation of DCs to further activate T cells.

### NIL-IM-Lip enhanced the killing ability of NK cells in vitro

NK cells are one critical tumour cell killer. The killing effects of NK cells were evaluated on B16F10 cells by MTT assay (Fig. 5k). Isolating high-purity NK cells from mice is a prerequisite for detecting NK cell toxicity. The magnetic-activated cell sorting (MACS) system is a mature cell separation method that causes little damage and yields high-purity samples. NK cells were separated from the spleens of C57BL/6 female mice by the MACS system and the NK cells were marked as CD3⁻CD49⁺. The positive rate of CD3⁻CD49b⁺ NK cells was 81.9% as determined by flow cytometric analysis, which could be used in the subsequent experiments (Supplementary Fig. 16). The cytotoxicity of NK cells with 20:1 target ratio was stronger than cells with 10:1 target ratio. Furthermore, preincubation of NIL-IM-Lip with the MMP2 enzyme showed similar cytotoxicity to that in the IL-2 and IL-15 groups (*P* > 0.05) (Fig. 5l). The above results indicated that the

conjugation of IL-15 onto the surface of NIL-IM-Lip did not affect the NK stimulation of IL-15.

### NIL-IM-Lip+L showed great antitumour efficacy in B16F10 primary tumour model

The antitumour efficacy of NIL-IM-Lip+L in vivo was evaluated in B16F10 tumour-bearing female C57BL/6 mice. The schedule of the in vivo administration approach is shown in Fig. 6a. We set up 14 groups (1) Normal Saline (NS), (2) Blank-Lip, (3) IR780, (4) IR780 + L, (5) 1-MT, (6) IL-15, (7) N-I-Lip+L, (8) IR780 + 1-MT + L, (9) NGR-IR780/1-MT-Lip+Laser (N-IM-Lip+L), (10) IR780 + IL-15 + L (IR780 + IL-15+Laser), (11) NIL-I-Lip+L, (12) IR780 + 1-MT + IL-15+Laser (IR780 + 1-MT + IL-15 + L), (13) IL-15-IR780/1-MT-Lip+Laser (IL-IM-Lip+L), and (14) NIL-IM-Lip+L. Treatment with NIL-IM-Lip+L gave tumours with smaller volumes than IR780 + 1-MT + IL-15 + L (*P* < 0.001) and IL-IM-Lip+L (*P* < 0.01) treatment (Fig. 6b and Supplementary Figs. 17 and 18). Tumour image was shown in Fig. 6c, and the tumour weights in the NIL-IM-Lip+L group were the lowest compared with those in the other groups (*P* < 0.001) (Fig. 6d). The body weights of the mice were not significantly changed during the experimental period (Fig. 6e). NIL-IM-Lip+L group showed stronger antitumour efficacy and the mice of the NIL-IM-Lip+L group had a longer survival rate than that in the other groups (Fig. 6f). Furthermore, the antitumour efficacy of NIL-IM-Lip without laser was also evaluated. The mice in the NIL-IM-Lip group showed larger tumour

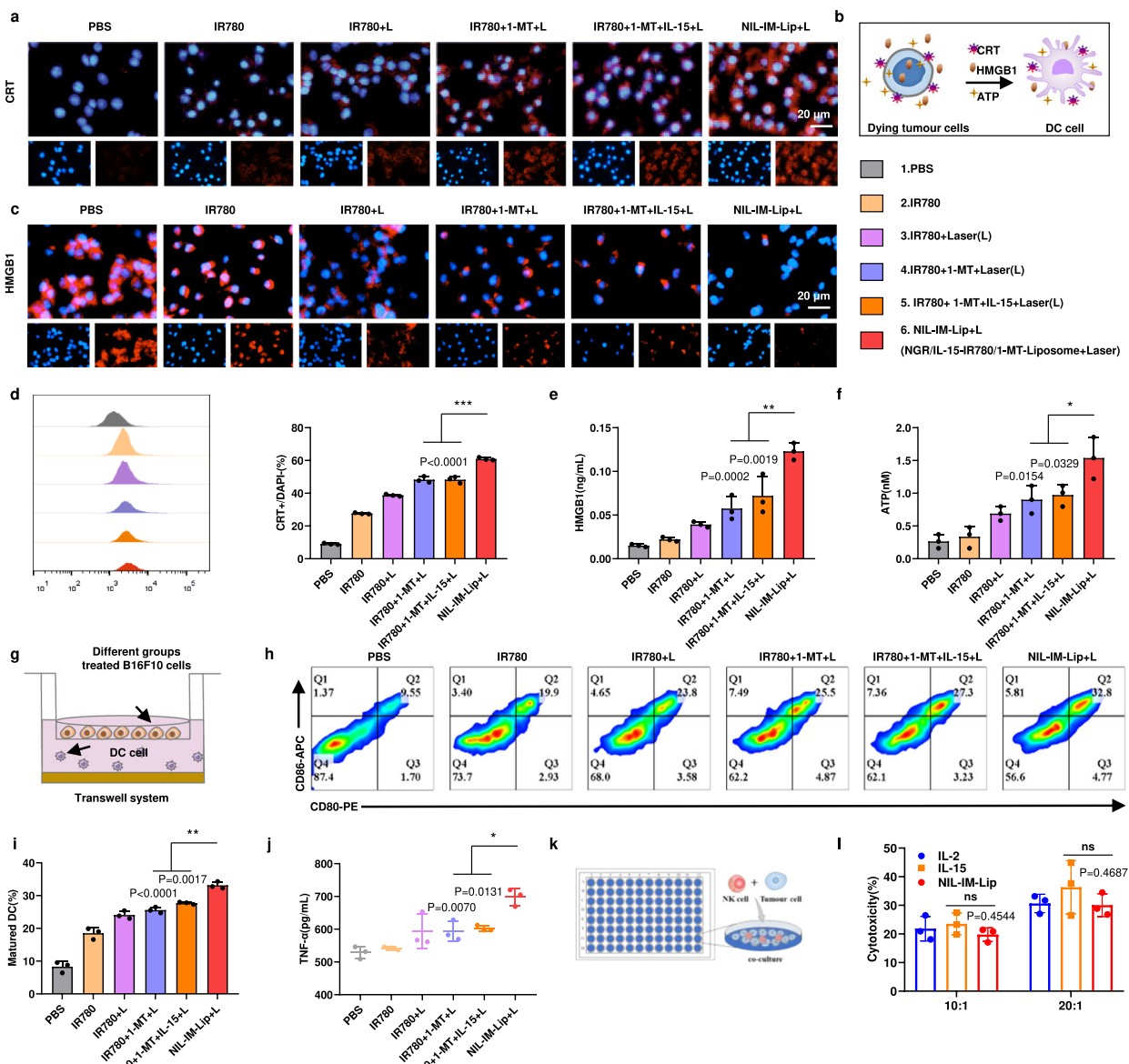

**Fig. 5 | The properties of NIL-IM-Lip + L on the ICD effect, DC maturation and the killing effects of NK cells in vitro. a** Fluorescence microscopy images of CRT exposure to B16F10 cells. The IR780 + L, IR780 + 1-MT + L, IR780 + 1-MT + IL-15 + L and NIL-IM-Lip+L groups were irradiated with an 808 nm laser for 5 min (1.0 W/cm²) (*n* = 3 biologically independent experiments). **b** Schematic illustration of the ICD effect to induce tumour cell death, including CRT exposure, HMGB1 release, and ATP secretion, which could promote the maturation of DCs. **c** Fluorescence microscopy images of HMGB1 release from B16F10 cells. The IR780 + L, IR780 + 1-MT + L, IR780 + 1-MT + IL-15 + L and NIL-IM-Lip+L groups were irradiated with an 808 nm laser for 5 min (1.0 W/cm²) (*n* = 3 biologically independent experiments). **d** Flow cytometric analysis of CRT exposure to B16F10 cells (*n* = 3 biologically independent experiments). **e** Quantitative ELISA analysis of HMGB1 release into the B16F10 culture medium supernatant (*n* = 3 biologically independent experiments).

**f** Quantitative ELISA analysis of ATP secretion into the B16F10 culture medium supernatant (*n* = 3 biologically independent experiments). **g** Schematic illustration of the transwell system used for DC maturation detection. **h, i,** Flow cytometric analysis of the ratio of mature DCs after coculture with B16F10 cells prestimulated with different formulations (*n* = 3 biologically independent experiments). **j** Quantitative ELISA analysis of TNF-α secretion in the DC culture medium supernatant (*n* = 3 biologically independent experiments). **k** Schematic illustration of the experimental process to determine the killing ability of NK cells. **l,** B16F10 cells killed by NK cells after pretreatment with different formulations (*n* = 3 biologically independent experiments). Data are presented as mean values ± SD. Statistical significance was calculated by the one-way ANOVA analysis of variance with Tukey's post hoc test. *P < 0.05, **P < 0.01, ***P < 0.001, ns P > 0.05. Source data are provided as a Source Data file. ns not significant.

volumes and greater tumour weights than the NIL-IM-Lip+L group, demonstrating that irradiation with the laser was vital for antitumour therapy (Supplementary Fig. 19). Proliferation and apoptosis of the tumour tissues were evaluated by H&E, Ki67, and TUNEL staining. As shown in the Fig. 6g and Supplementary Fig. 20, the NIL-IM-Lip+L group showed more nuclear shrinkage by H&E staining, fewer Ki67-positive cells, and more TUNEL-positive cells, which indicated that NIL-IM-Lip+L could effectively inhibit the proliferation and enhance the apoptosis of tumour cells.

Furthermore, the safety experiment was preliminarily evaluated by conducting a haemolysis test and staining the main organs with H&E. There was no obvious haemolysis at concentrations of 0.5–10 μg/mL NIL-IM-Lip+L, as the haemolysis rates at these concentrations were less than 5% (Supplementary Fig. 21). Moreover, after 16 days, the mice were euthanized and the major organs (including the heart, liver, spleen, lung, and kidneys) were taken out. The H&E staining results showed no visual pathological changes in any group (Supplementary Fig. 22). All of the above results preliminarily indicated that NIL-IM-Lip was safe.

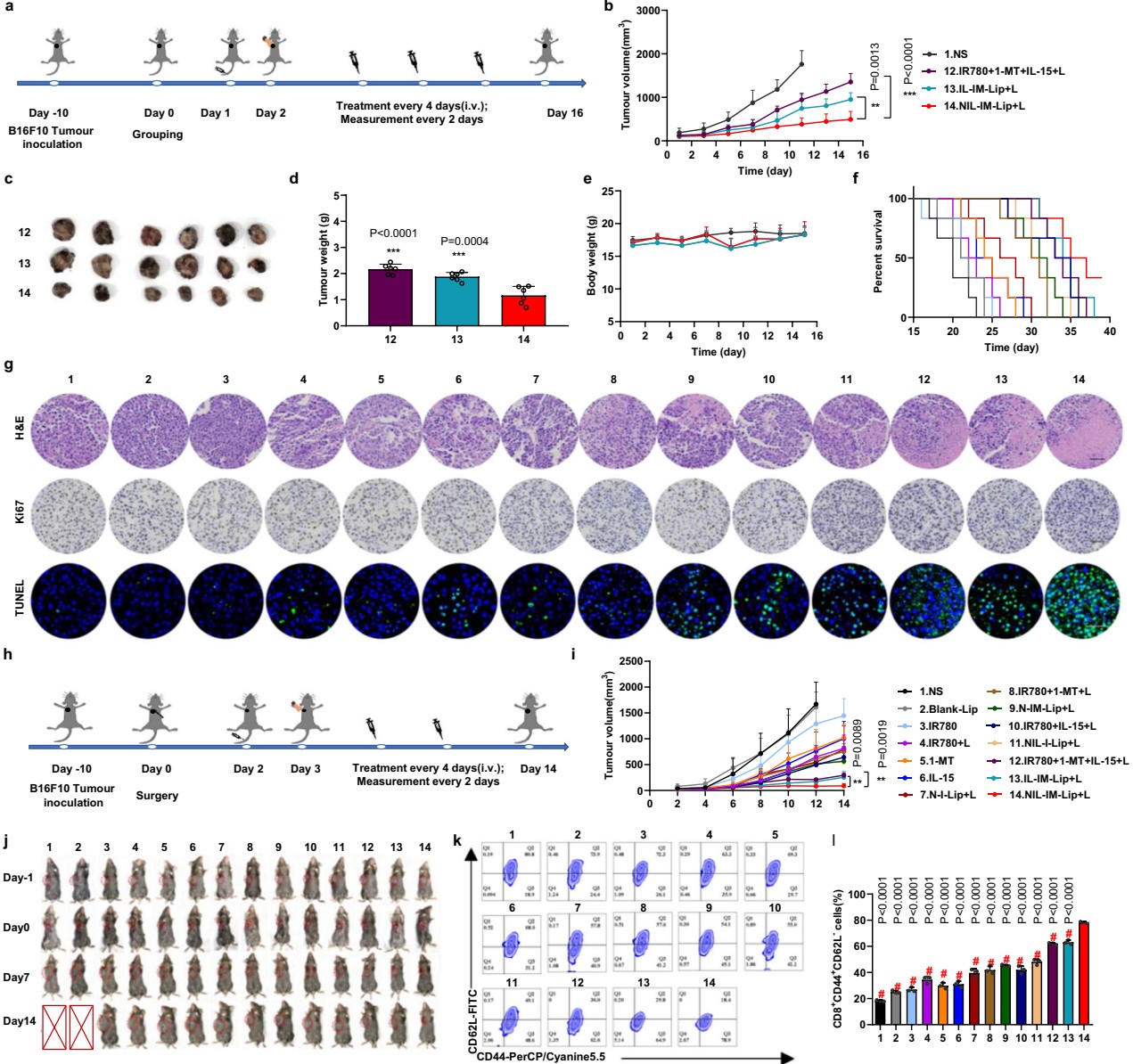

**Fig. 6 | The antitumour and postsurgical recurrence efficacy of NIL-IM-Lip + L in the B16F10 primary and postsurgical recurrence models.** Mice were euthanized when the tumour volumes reached ~2000 mm³. **a** Schedule of the antitumour experiment in B16F10 tumour-bearing C57BL/6 mice. **b** Tumour volume curves of NS, IR780 + 1-MT + IL-15 + L, IL-IM-Lip, and NIL-IM-Lip+L treated B16F10 tumour-bearing C57BL/6 mice (n = 6 biologically independent animals per group). **c** Photographs of tumours after 16 days of treatment (n = 6 biologically independent animals per group). **d** Tumour weights in different formulation-treated groups (n = 6 biologically independent animals per group). **e** The body weights of different formulation-treated mice (n = 6 biologically independent animals per group). **f** Percent survival of different formulation-treated B16F10 tumour-bearing C57BL/6 mice (n = 6 biologically independent animals per group). **g** Immunohistochemical images of tumour tissue sections (n = 3 biologically

independent experiments). Scale bar:50 µm. **h** Schedule of the antitumour post-surgical recurrence experiment on B16F10 tumour-bearing C57BL/6 mice. **i** Tumour volume curves of different formulation-treated B16F10 tumour-bearing C57BL/6 mice (n = 5 biologically independent animals per group). **j** Images of mice during treatment in the antitumour postsurgical recurrence experiment. **k, l** Flow cytometric analysis of memory T cells from the spleens of mice after different treatments (n = 3 biologically independent experiments per group). Data are presented as mean values ± SD. Statistical significances were calculated by one-way analysis of variance with Tukey's post hoc test. **P < 0.01, ***P < 0.001, #P < 0.001. The statistical significances of **b** were calculated of the (12) (IR780 + 1-MT + IL-15 + L, (13) IL-IM-Lip+L, and (14) NIL-IM-Lip+L groups on day 15. Source data are provided as a Source Data file. N-IM-Lip+L NGR-IR780/1-MT-Lip+Laser NIL-I-Lip+L NGR/IL-15-IR780-Lip+Laser, IL-IM-Lip+L IL-15-IR780/1-MT-Lip+Laser.

## NIL-IM-Lip+L showed great postsurgical recurrence efficacy and strong immune memory

Surgical resection is the preferred clinical method for the treatment of melanoma. Hence, the postsurgical recurrence models were used to evaluate the antitumour efficacy (Fig. 6h, Supplementary Fig. 23). Similar to the antitumour efficacy in the primary B16F10 model, NIL-IM-Lip+L greatly inhibited tumour growth (Fig. 6i). The tumour volume curves of each mouse in every group were shown in Supplementary

Fig. 24. Moreover, the NIL-IM-Lip without irradiation of laser showed lower antitumour efficacy than NIL-IM-Lip+L (Supplementary Fig. 25). The images of mice at -1, 0, 7 and 14 days showed consistent changes in the tumour volume curves (Fig. 6j). After 14 days, the mice were euthanized and the spleens were taken out to evaluate memory T cells. Memory T cells were marked as CD3⁺CD8⁺CD44⁺CD62L⁻. The frequency of memory T cells in the NIL-IM-Lip+L group was 4.3-fold higher than that in the NS group (P < 0.001) (Fig. 6k, l).

### NIL-IM-Lip+L coactivated CTLs and NK cells and remodelled the suppressive TLIME

The frequencies of immune cells (CD4$^+$ T cells, CD8$^+$ T cell, Treg cells, NK cells and CTLs) and the concentrations of cytokines (including IL-6, IL-12, Interferon (IFN)-γ, IL-10 and TGF-β) in tumour tissues were used to evaluate the immune status of the tumour microenvironment. Flow cytometry analysis revealed that the intratumoural frequencies of CD4$^+$ T and CD8$^+$ T cells in the NIL-IM-Lip+L group were 3.7-fold and 7.8-fold higher than those in the NS group, respectively ($P < 0.001$, $P < 0.001$). The CD8$^+$ T cells in the NIL-I-Lip+L were higher compared with N-I-Lip+L ($P < 0.01$), revealing the modification of IL-15 on the liposome still activated more T cells (Fig. 7a–c). Besides, the intratumoural frequencies of CTLs (IFN-γ$^+$CD8$^+$ T cells) in NIL-IM-Lip+L group was higher than IR780 + 1-MT + IL-15 + L group and IL-IM-Lip+L groups (Fig. 7d). The level of NK cells in the NIL-I-Lip+L group was higher than that in the N-I-Lip+L group, indicating that IL-15 could effectively activate NK cells ($P < 0.001$). The NIL-IM-Lip+L group displayed the highest abundance of NK cells compared with the other groups ($P < 0.001$) (Fig. 7e, f). Additionally, the 1-MT greatly decreased the Treg cells in the tumour compared with NS group, indicating the IDO1 inhibitor indeed inhibited the infiltration of Treg cells. The frequency of Treg cells in the NIL-IM-Lip+L group was lower compared with the 1-MT group and IR780 + 1-MT + IL-15 + L group ($P < 0.001$, $P < 0.01$) (Fig. 7g). Considering the nonspecific distribution of free drug and the tumour and LNs accumulation of NIL-IM-Lip, we focused on comparing the immune cell changes between the IR780 + 1-MT + IL-15 + L group and the NIL-IM-Lip+L group. Significantly, the NIL-IM-Lip+L group had more CD8$^+$ T cells, NK cells, and CTLs and fewer Treg cells in the tumour tissues than the IR780 + 1-MT + IL-15 + L group, showing better immune cell regulation in tumour tissues. Moreover, the ratio of kynurenine (Kyn) and tryptophan (Trp) was studied (Supplementary Fig. 26). The IR780 + L and N-I-Lip+L groups greatly enhanced the ratio of Kyn/Trp compared with the NS group ($P < 0.05$, $P < 0.001$), which was attributed to the overexpression of IDO1. The 1-MT, IR780 + 1-MT + L, N-IM-Lip+L and NIL-IM-Lip+L groups effectively decreased the ratio of Kyn/Trp compared with the NS, IR780 + L and N-I-Lip+L groups ($P < 0.001$), demonstrating that 1-MT could strongly inhibit IDO1 functions. The NIL-IM-Lip+L group generated more IL-6, IL-12 and IFN-γ than the other groups, but the opposite trend was observed for IL-10 and TGF-β (Fig. 7h and Supplementary Fig. 27). The ICD effect induced by NIL-IM-Lip+L was further evaluated in vivo. The NIL-IM-Lip+L group showed higher CRT exposure and less HMGB1 in the nucleus (Supplementary Figs. 28, 29), which was consistent with the in vitro data.

Furthermore, the LNs microenvironment in vivo was further evaluated. The immune cells in LNs (including DCs, NK cells, CD8$^+$ T cells and Treg cells) were also studied. As shown in Fig. 7i–k, the frequency of CD4$^+$ T and CD8$^+$ T cells in the NIL-IM-Lip+L group were 2.4-fold and 2.5-fold higher than those in the NS group ($P < 0.001$, $P < 0.001$). Similar with the results in the tumour, IL-15 could also effectively activate the NK cells in the LNs. The NIL-I-Lip+L enhanced the infiltration of NK cells in the LNs compared with N-I-Lip+L ($P < 0.001$). The NIL-IM-Lip+L group showed the highest abundance of NK cells than other groups (Fig. 7l, m). Also, the 1-MT greatly decreased the infiltration of Treg cells in the LNs than NS group ($P < 0.001$). The number of Treg cells in the NIL-IM-Lip+L group was 24.4-fold lower than that in the NS group ($P < 0.001$) (Fig. 7n). Finally, the maturity of DCs was evaluated in vivo, and NIL-IM-Lip+L exhibited the strongest ability to produce mature DCs (Fig. 7o, p). All of these results proved that the NIL-IM-Lip directed to the LNs could also reverse the suppressive LNs microenvironment. Similar to tumour tissues, we focused on comparing the immune cell changes between the IR780 + 1-MT + IL-15 + L group and NIL-IM-Lip group. Notably, NIL-IM-Lip+L treatment showed better immune cell regulation, including activating more CD8$^+$ T cells, NK cells, and fewer Treg cells in LNs.

In summary, the immune cell changes in the TLIME indicated that NIL-IM-Lip+L could coactivate T cells and NK cells and inhibit Treg cells in the TLIME to reverse the suppressive TLIME. Notably, compared with the IR780 + 1-MT + IL-15 + L group, the NIL-IM-Lip+L group exhibited better immune cell regulation in the TLIME, verifying that the pH/MMP2/temperature triple-sensitive immunomodulatory nanoinducer NIL-IM-Lip could accumulate in tumour tissues and be directed to the LNs.

### NIL-IM-Lip+L combined with the PD-1 mAb showed enhanced antitumour effects in hot tumours (B16F10 model) and cold tumours (CT26 model)

To further evaluate the application of NIL-IM-Lip+L, the combined therapy of NIL-IM-Lip+L and PD-1 mAb was investigated on the B16F10 model (hot tumour). A total of $8 \times 10^5$ B16F10 tumour cells were subcutaneously inoculated into the right sides of female C57BL/6 mice. The schedule of the in vivo administration approach is shown in Fig. 8a. Smaller tumour volumes were observed after NIL-IM-Lip+PD-1 mAb +Laser (NIL-IM-Lip+PD-1+L) treatment than those in the PD-1 group and NIL-IM-Lip+L group ($P < 0.001$, $P < 0.001$) (Fig. 8b–d). Notably, one mouse of NIL-IM-Lip+PD-1+L were cured and the NIL-IM-Lip+PD-1 + L gave a 20% complete response (CR), which indicated the great potential of NIL-IM-Lip combined with the PD-1 mAb. Finally, the tumours were taken out and the weights were measured, giving results similar to those found from the tumour volume analysis (Fig. 8e). In the tumour tissues (Fig. 8f, g), the frequency of positive immune responders in the NIL-IM-Lip+PD-1+L group, including CD4$^+$ T cells, CD8$^+$ T cells, CTLs and NK cells, were higher than those in the NIL-IM-Lip+L group ($P < 0.001$, $P < 0.001$, $P < 0.05$, $P < 0.001$) and PD-1 group ($P < 0.001$, $P < 0.001$, $P < 0.001$, $P < 0.001$). The NIL-IM-Lip+PD-1+L group greatly suppressed negative Treg cells compared with NIL-IM-Lip+L group and PD-1 group ($P < 0.001$) (Fig. 8f). Consistent with changes in the immune state in the tumour tissues, NIL-IM-Lip+PD-1+L activated more CD4$^+$ T cells, CD8$^+$ T cells and NK cells and inhibited more Treg cells in the LNs (Fig. 8h, i). These results confirmed that NIL-IM-Lip+PD-1+L reversed the suppressive TLIME and established antitumour immunity. Besides, the combined therapy of NIL-IM-Lip+L and PD-1 mAb was further investigated on the cold tumour (CT26 model). The in vitro assay of cell apoptosis and CRT exposure were evaluated on the CT26 and MC38 cells. The results showed that the NIL-IM-Lip+L had stronger cells apoptosis rate and more CRT exposure on both CT26 and MC38 cells (Supplementary Figs. 30–33). Then the in vivo antitumour effects were evaluated on CT26 tumour model. In BALB/c mice bearing CT26 colorectal cancer tumours (Fig. 8j), NIL-IM-Lip+PD-1+L showed stronger antitumour efficacy and a similar immune response to those observed in the B16F10 model (Fig. 8k–r). Finally, the longer survival time of the mice in the NIL-IM-Lip+PD-1+L group on both B16F10 model and CT26 model further demonstrated the advantages of the combination therapy (Supplementary Fig. 34). These results verified that NIL-IM-Lip combined with PD-1 mAb showed great antitumour immunotherapy in both hot tumours and cold tumours by remodelling the suppressive TLIME and simultaneously comobilizing T and NK cells in the TLIME.

## Discussion

Given the importance of communication between tumours and LNs during immunotherapy, we proposed that remodelling the suppressive TLIME while simultaneously comobilizing CTLs and NK cells in the TLIME could produce a stronger antitumour effect. pH/MMP2/temperature triple-sensitive NIL-IM-Lip was elaborately designed for targeted tumour delivery and direction to the LNs. The accumulated NIL-IM-Lip in the TLIME initiated the immune response and simultaneously coactivated CTLs and NK cells. Moreover, the released 1-MT relieved Treg cell infiltration in the TLIME and remodelled the suppressive TLIME. NIL-IM-Lip combined with a PD-1 mAb showed robust

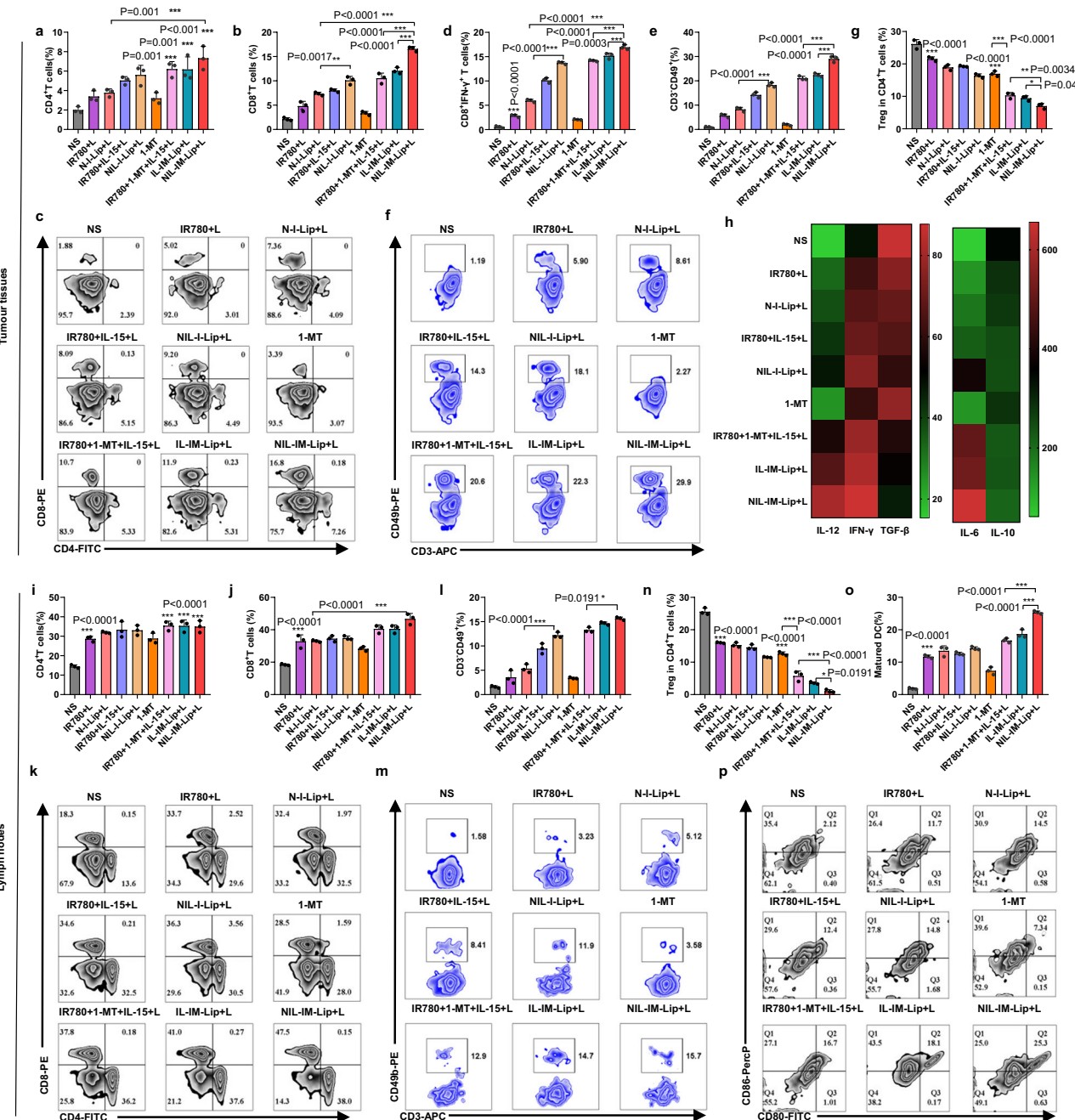

**Fig. 7 | Effects of NIL-IM-Lip + L treatment on the simultaneous coactivation of CTLs and NK cells and the reversion of the suppressive conditions in the TLIME in the B16F10 model. a–c** Flow cytometric analysis of intratumoural CD4+ T cells and CD8+ T cells. **d** Flow cytometric analysis of the infiltration of CTLs into tumours (*n* = 3 biologically independent experiments per group). **e, f** Flow cytometric analysis of NK cell infiltration into the tumours (*n* = 3 biologically independent experiments per group). **g** Flow cytometric analysis of the infiltration of Treg cells gated on CD4+ lymphocytes in the tumours (*n* = 3 biologically independent experiments per group). **h** Quantitative ELISA analysis of the cytokines in the tumour tissues from mice treated with different formulations (*n* = 3 biologically independent experiments per group). **i–k** Flow cytometric analysis of the infiltration of CD4+ T cells and CD8+ T cells in the LNs (*n* = 3 biologically independent experiments per group). **l, m** Flow cytometric analysis of NK cell infiltration into LNs (*n* = 3 biologically independent experiments per group). **n** Flow cytometric analysis of the infiltration of Treg cells gated on CD4+ lymphocytes in the LNs (*n* = 3 biologically independent experiments per group). **o, p** Flow cytometric analysis of DC infiltration in LNs (*n* = 3 biologically independent experiments per group). Data are presented as mean values ± SD. Statistical significances were calculated by one-way analysis of variance with Tukey's post hoc test. \**P* < 0.05, \*\**P* < 0.01, \*\*\**P* < 0.001. Source data are provided as a Source Data file. N-I-Lip+L NGR-IR780-Lip+Laser, NIL-I-Lip+L NGR/IL-15-IR780-Lip+Laser, IL-IM-Lip+L IL-15-IR780/1-MT-Lip+Laser.

antitumour effects in hot tumours (B16F10 model) and cold tumours (CT26 model). Notably, a 20% complete response was observed in the B16F10 model. Our work provides a platform that can deliver both NIL-IM-Lip and a PD-1 mAb into hot tumours, cold tumours and LNs to enhance immunotherapy, which greatly promotes the efficacy of the PD-1 mAb and shows great clinical application prospects.

## Methods

### Animals

Female C57BL/6 mice (6-8 weeks) were purchased from SPF Biotechnology Co., Ltd. (Beijing) (Stock number: B204). Female BALB/c mice (6–8 weeks) were purchased from Beijing Vital River Laboratory Animal Technology Co., Ltd (Stock number: 211). Mice were housed

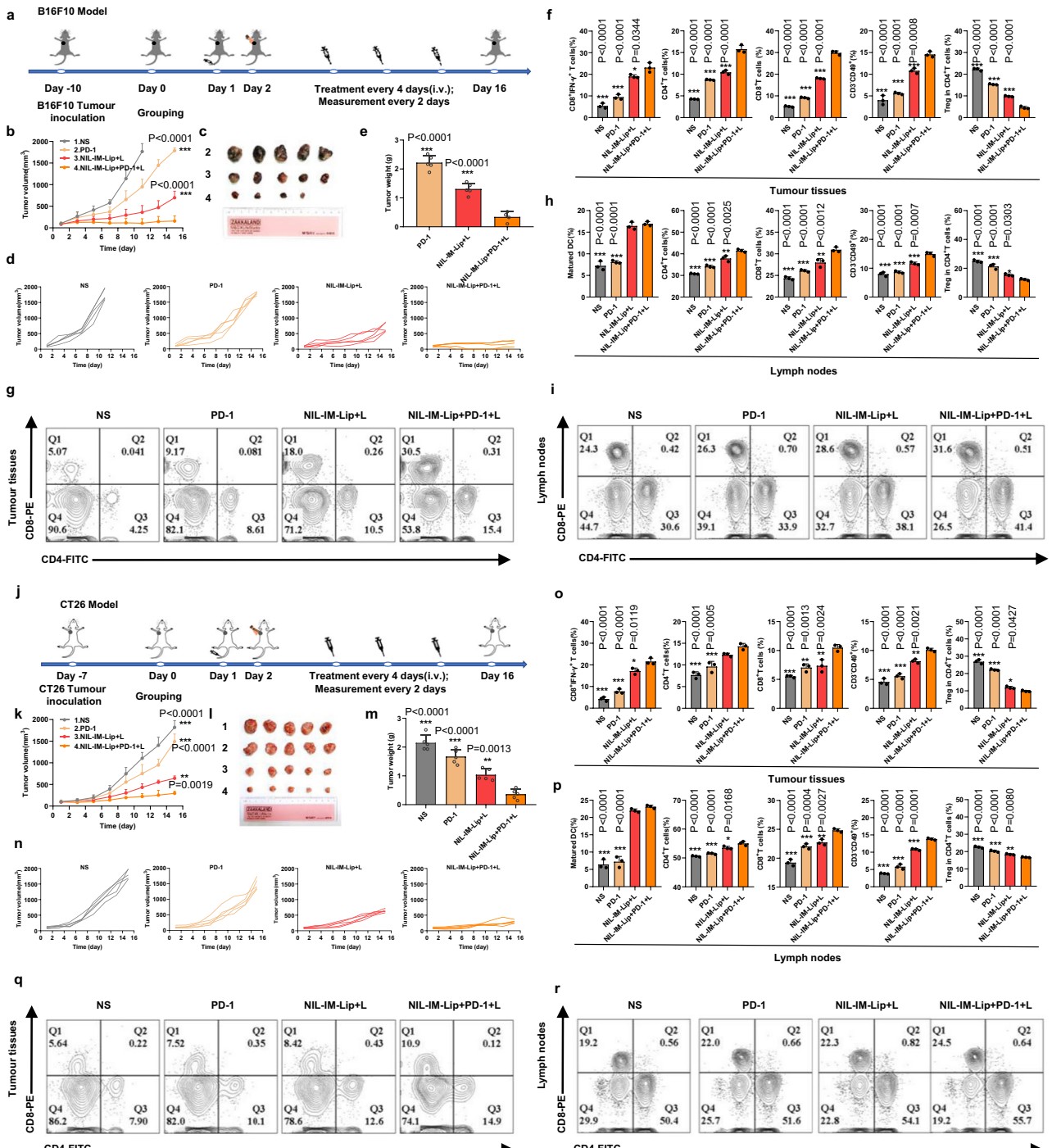

**Fig. 8 | Amplified antitumour effects of NIL-IM-Lip + L and PD−1 mAb cotreatment in hot tumours (B16F10 model) and cold tumours (CT26 model).** Mice were euthanized when the tumour volumes reached ~2000 mm³. **a** Schedule of the NIL-IM-Lip+L combined with PD-1 mAb experiment in B16F10 tumour-bearing C57BL/6 mice. **b** Tumour volume curves of different formulations (*n* = 5 biologically independent animals per group). **c** Photographs of tumours after 16 days of treatment (*n* = 5 biologically independent animals per group). **d** Individual tumour volume curves after treatment with different formulations (*n* = 5 biologically independent animals per group). **e** Tumour weights in different formulation-treated groups (*n* = 5 biologically independent animals per group). **f, g** Flow cytometric analysis of the intratumoural CTLs, CD4⁺ T cells and CD8⁺ T cells, NK cells and Treg cells (*n* = 3 biologically independent experiments per group). **h, i** Flow cytometric analysis of the infiltration of DCs, CD4⁺ T cells, CD8⁺ T cells, NK cells and Treg cells in the LNs (*n* = 3 biologically independent experiments per group). **j** Schedule of the NIL-IM-Lip+L combined with PD-1 mAb antitumour

experiment in CT26 tumour-bearing BALB/c mice. **k** Tumour volume curves of different formulations (*n* = 5 biologically independent animals per group). **l**, Photographs of tumours after 16 days of treatment (*n* = 5 biologically independent animals per group). **m** Tumour weights in different formulation-treated groups (*n* = 5 biologically independent animals per group). **n** Individual tumour volume curves after treatment with different formulations (*n* = 5 biologically independent animals per group). **o, q** Flow cytometric analysis of the intratumoural CTLs, CD4⁺ T cells, CD8⁺ T cells **q**, NK cells and Treg cells (*n* = 3 biologically independent experiments per group). **p, r** Flow cytometric analysis of the infiltration of DCs, CD4⁺ T cells and CD8⁺ T cells (r), NK cells and Treg cells in the LNs (*n* = 3 biologically independent experiments per group). Data are presented as mean values ± SD. Statistical significances were calculated by one-way analysis of variance with Tukey's post hoc test. \**P* < 0.05, \*\**P* < 0.01, \*\*\**P* < 0.001. Source data are provided as a Source Data file. PD-1: PD-1 mAb; NIL-IM-Lip+PD-1 + L: the combination the NIL-IM-Lip+L and PD-1 mAb.

under conditions of a light/dark cycle of 12 h, an ambient temperature of $25 \pm 2\,°C$, and a humidity of $60 \pm 10\%$. All the animals were treated according to the Guide for Care and all relevant animal experiments were performed in compliance with the animal management rules of the Ministry of Health, People's Republic of China and Animal Experiment Ethics Review Board of Shandong University. When the tumour volumes reached ~2000 mm$^3$, the mice were euthanized by rapidly dislocation of the head and neck.

## Materials

1,2-Dipalmitoyl-sn-glycero-3-phosphocholine (DPPC) and Hydrogenated Soybean Phospholipid (HSPC) was purchased from AVT Pharmaceutical Tech Co. (Shanghai). DSPE-Hyd-PEG$_{5000}$-MAL and DSPE-PEG$_{2000}$-NH$_2$ were obtained from Xi'an Ruixi Biological Technology Co., Ltd. 1-MT was purchased from Sigma–Aldrich (United States). IR780 was obtained from Shanghai Aladdin Bio-Chem Technology Co., Ltd. NGR was purchased from Leon Biological Technology Co. Ltd. (Nanjing, China). MMP2 sensitive/non-sensitive peptides were purchased from Nanjing Peptide Biological Science and Technology Limited Co., Ltd. (Nanjing, China). Murine IL-2, recombinant murine granulocyte-macrophage colony-stimulating factor (GM-CSF), murine IL-4 and murine IL-15 were purchased from Pepro Tech (United States). Collagenase type IV (MMP2 enzyme) was purchased from Sigma–Aldrich (San Diego, USA). An Annexin V-FITC/PI apoptosis detection kit, the BCA protein assay kit and ATP assay kit were purchased from Beyotime Biotechnology Co., Ltd. (Shanghai, China). HMGB1 ELISA kits and Methylthiazol tetrazolium (MTT) was purchased from Solarbio (Shanghai, China). IFN-γ, TNF-α, TGF-β, IL-10, IL-6 and IL-12 ELISA kits were purchased from MultiSciences (Lianke) Biotech Co., Ltd. The PD-1 antibody, MojoSort™ Mouse NK Cell Isolation Kit protocol, MojoSort™ Buffer (5X) and MojoSort™ Magnet were purchased from Dakewei Co., Ltd. (Shenzhen, China). All other materials used were of analytical reagent grade.

## Antibodies

APC anti-mouse CD3 Antibody (BioLegend; Catalog number: 100236; Clone name: 17A2; 1:100 dilution), FITC anti-mouse CD4 Antibody (BioLegend; Catalog number: 100405; Clone name: GK1.5; 1:400 dilution), PE anti-mouse CD8a Antibody (BioLegend; Catalog number: 100708; Clone name: 53–6.7; 1:200 dilution), PE anti-mouse CD4 Antibody (BioLegend; Catalog number: 100407; Clone name: GK1.5; 1:200 dilution), Alexa Fluor® 647 anti-mouse FOXP3 Antibody (BioLegend; Catalog number: 126408; Clone name: MF-14; 1:100 dilution), FITC anti-mouse CD62L Antibody (BioLegend; Catalog number: 104405; Clone name: MEL-14; 1:400 dilution), PerCP/Cyanine5.5 anti-mouse/human CD44 Antibody (BioLegend; Catalog number: 103031; Clone name: IM7; 1:200 dilution), PE anti-mouse CD49b (pan-NK cells) Antibody (BioLegend; Catalog number: 108907; Clone name: DX5; 1:200 dilution), APC anti-mouse IFN-γ Antibody (BioLegend; Catalog number: 505810; Clone name: XMG1.2; 1:50 dilution), FITC anti-mouse CD11c Antibody (BioLegend; Catalog number: 117306; Clone name:N418; 1:400 dilution), PE anti-mouse CD80 Antibody (BioLegend; Catalog number: 104707; Clone name: 16-10A1; 1:100 dilution), APC anti-mouse CD86 Antibody (BioLegend; Catalog number: 105011; Clone name: GL-1; 1:200 dilution), PE anti-mouse CD11c Antibody (BioLegend; Catalog number: 117308; Clone name:N418; 1:200 dilution), FITC anti-mouse CD80 Antibody (BioLegend; Catalog number: 104706; Clone name: 16-10A1; 1:100 dilution), PerCP/Cyanine5.5 anti-mouse CD86 Antibody (BioLegend; Catalog number: 105028; Clone name: GL-1; 1:50 dilution). The Calreticulin Polyclonal Antibody (Bioss; Catalog: bs-5913R; 1:200 dilution), Rabbit Anti-HMGB1 antibody (Bioss; Catalog:bs-0664R; 1:200 dilution) and Goat Anti-Rabbit IgG/Alexa Fluor 594 antibody (Bioss; Catalog:bs-0295G-AF594; 1:400 dilution) and Goat Anti-Rabbit IgG/Alexa Fluor 488 antibody (Bioss; Catalog:bs-0295G-AF488; 1:400 dilution) were purchased from Bioss.

## Cell lines

Mouse malignant melanoma cells (B16F10), mouse colorectal cancer cells (CT26, MC38) and human umbilical vein endothelial cells (HUVECs) obtained from the Chinese Academy of Sciences (China, Shanghai Institutes for Biological Sciences). B16F10 cells, CT26 cells and NK cells isolated from C57BL/6 female mice were incubated in RPMI 1640 media with streptomycin and penicillin (1%) and 10% fetal bovine serum (FBS). MC38 cells and HUVECs were incubated in DMEM with streptomycin and penicillin (1%) and FBS (10%). All cells were cultured in a 37 °C incubator with 5% CO$_2$. All the cell lines showed negative for mycoplasma contamination.

## Preparation and characterization of NIL-IM-Lip

DSPE-PEG$_{2000}$-MMP2 sensitive/non-sensitive pep-Mal were synthesized as shown in Supplementary Fig. 1. Mal-MMP2 sensitive/non-sensitive pep-COOH was dissolved in dimethyl formamide (DMF) and activated with 1-ethyl-3-(3-dimethylaminopropyl) carbodiimide (EDC) and N-hydroxysuccinimide (NHS) in an ice bath for 1 h. Then, DSPE-PEG$_{2000}$-NH$_2$ was dissolved in DMF, and the two reaction solutions were mixed and stirred for 12 h. The obtained solutions were purified via dialysis against deionized water (MWCO = 3500 Da) for 48 h. DSPE-Hyd-PEG$_{5000}$-NGR was synthesized as shown in Supplementary Fig. 3. Briefly, NGR and DSPE-Hyd-PEG$_{5000}$-Mal with a molar ratio of 1.5:1 were dissolved in PBS, pH 7.4, containing 1 mM EDTA. Then, the solution was stirred at room temperature under nitrogen protection for 12 h. The excess NGR was removed via dialysis against deionized water (MWCO = 3500 Da) for 48 h, followed by lyophilization. The structures of DSPE-Hyd-PEG$_{5000}$-NGR and DSPE-PEG$_{2000}$-MMP2-sensitive/non-sensitive pep-Mal were verified by $^1$H NMR. Thiolation of IL-15 was carried out with Traut's reagent. IL-15 was diluted in PBS, pH 7.8-8.0, containing 5 mM EDTA, a 20-fold molar excess of Traut's reagent was added, and the solutions were mixed. Then, the solutions were stirred for 1 h at room temperature. Excess Traut's reagent was removed by using a Thermo Scientific Zeba column. After that, IL-15-SH and DSPE-PEG$_{2000}$-MMP2-sensitive pep-Mal were dissolved in PBS (pH 7.4 with 1 mM EDTA) and the solution stirred at room temperature under nitrogen protection for 12 h.

NIL-IM-Lip was prepared by the film dispersion method. Briefly, DPPC (10 mg/mL), cholesterol (1.25 mg/mL), and IR780 (0.2 mg/mL) were dissolved in ethanol and evaporated to form a dried lipid film (40 °C, 30 min). Then, the dried lipid film was hydrated with 1-MT (1.25 mg/mL) containing DSPE-Hyd-PEG$_{5000}$-NGR (3% mol ratio relative) and DSPE-PEG$_{2000}$-MMP2-sensitive pep-IL-15. DPPC liposomes (Blank-Lip) and liposomes containing IR780 and 1-MT (IM-Lip) were prepared as described above. The large liposomes (NIL-IM-Lip-Large, NIL-IM-Lip-L) were extruded five times using a LiposoFast-Basic extruder (100 nm) (Avestin, Ottawa, ON, Canada), and the small liposomes (NIL-IM-Lip-Small, NIL-IM-Lip-S) were extruded using a 50 nm membrane. A Malvern Zeta Sizer Nano-ZS instrument was selected to detect the particle size, polydispersity index (PDI) and zeta potential of NIL-IM-Lip. Transmission electronic microscopy (TEM) was selected to visualize the morphology of NIL-IM-Lip. The encapsulation efficiency (EE%) and drug loading (DL%) of NIL-IM-Lip were calculated by the following equations:

$$DL\% = W_{loaded\,drug}/W_{liposome} \times 100\% \qquad (1)$$

$$EE\% = W_{loaded\,drug}/W_{total\,drug} \times 100\% \qquad (2)$$

where $W_{loaded\,drug}$ and $W_{liposome}$ represent the weight of drug in the liposome and weight of all added to the liposome. $W_{total\,drug}$ is the weight of drug added to the liposome.

To further verify the effective loading of IR780 and 1-MT, ultraviolet−visible (UV − vis) spectrophotometric analyses were performed.

Briefly, IR780, 1-MT, and NIL-IM-Lip were subjected to scanning from 200 nm to 900 nm.

### In vitro drug release

The dialysis bag method was selected to determine the 1-MT release profiles. Briefly, 1.0 mL of free 1-MT, NIL-IM-Lip (DPPC) and NIL-IM-Lip (HSPC) was added to a dialysis bag, which were then placed into a tube with 10 mL of pH 7.4 PBS. The tubes were incubated at 37 °C with a stirring speed of 100 rpm. The 1-MT + laser and NIL-IM-Lip (DPPC or HSPC) + laser groups were subjected to 5 min of irradiation with an 808 nm NIR laser at 1.0 W/cm$^2$ twice with a 10 min interval between the two treatments. A sample was then collected, and the volume was replaced with 10 mL of fresh PBS, pH 7.4, at predetermined time points. The concentration of released 1-MT was determined by high performance liquid chromatography (HPLC). Experiments were performed in triplicate.

### MMP2-sensitive enzymatic digestion assay

The enzymatic digestion method was selected to evaluate the properties of the MMP2-sensitive peptide using the MMP2 enzyme. Briefly, 1.0 mg/mL MMP2-sensitive peptide was incubated with MMP2 enzyme (0, 0.5, 1, 5, 10, and 50 μg/mL) in PBS, pH 7.4, for 24 h at 37 °C. Subsequently, the enzymatic cleavage time was explored. MMP2-sensitive peptide (1.0 mg/mL) was incubated with 50 μg/mL MMP2 enzyme at 37 °C for 5, 15, 30, 60 and 90 min. The digested fragments were identified by HPLC.

### Stability of the liposomes in different media

The formulations were diluted with 20% plasma and RPMI 1640 medium, and then we measured the particle size of NIL-IM-Lip after incubation for different lengths of time. In addition, the particle size of NIL-IM-Lip was recorded every day for 7 days to evaluate the long-term physical stability.

### In vitro and In vivo photothermal performance

To investigate the appropriate IR780 concentration and laser irradiation power, a series of IR780–concentration and irradiation power–temperature curves were constructed after measuring with a thermocouple needle. Briefly, 1 mL of water and a series of concentrations of I-Lip (0.1, 1, 10, 30 μg/mL) were irradiated with an 808 nm laser at 1.0 W/cm$^2$ for 10 min. In addition, 10 μg/mL I-Lip was irradiated with an 808 nm laser at 0.1, 0.6, 1.0 and 2.0 W/cm$^2$ for 10 min. At predetermined times, the temperature was recorded. Then, 1 mL of water, free IR780 and I-Lip (10 μg/mL) were irradiated with an 808 nm laser at 1.0 W/cm$^2$ for 10 min, and we captured photographs using an infrared imaging device. IR780 and I-Lip were irradiated with an 808 nm laser for four on/off cycles, and the temperature was recorded to evaluate photothermal stability. The in vivo photothermal conversion efficiency was evaluated on female B16F10 tumour-bearing C57BL/6 mouse model. A total of $1 \times 10^6$ B16F10 cells (0.1 mL) were injected into the right axilla of C57BL/6 mice to establish tumour model. The mice were intravenously injected with NS, IR780, I-Lip or N-I-Lip (IR780, 1 mg/kg). The mice were then treated with laser irradiation (808 nm) at 1.0 W/cm$^2$ for 10 min 12 h after the administration of the different formulations. The mice were anaesthetized and imaged using an infrared thermal imaging camera.

### In vivo NIRF imaging

A female B16F10 tumour-bearing C57BL/6 mouse model was used to evaluate biodistribution. A total of $1 \times 10^6$ B16F10 cells (0.1 mL) were injected into the right axilla of C57BL/6 mice to establish tumour model. After 15 days, the mice were intravenously injected with free IR780, I-Lip or N-I-Lip (1.5 mg/kg) (n = 3 biologically independent animals per group). At 1, 2, 4, 8, 12, 24 and 48 h, mice were anaesthetized with gas and images were captured with a real-time IVIS spectrum.

Then, the mice were euthanized, and the tumours, LNs and major organs (including the heart, liver, spleen, lung, and kidneys) were taken out for imaging.

To evaluate the LNs accumulation of the different particle size nanoparticles and different mass ratio of cholesterol nanoparticles, $1 \times 10^6$ B16F10 cells (0.1 mL) was injected into the right axilla to establish the female C57BL/6 mice model. For evaluating the LNs accumulation of the different particle size nanoparticles, the mice were intravenously injected with a large nanoparticle (N-I-Lip-L) or a small nanoparticle (N-I-Lip-S) (1.5 mg/kg). At different time points (12, 24, 48 h), the mice were euthanized and the LNs were acquired for in vitro imaging. Then, the effect of cholesterol transport on the LNs accumulation capacity was studied. Nanoparticles consisting of 1/2, 1/8 and 1/16 mass ratios of cholesterol were prepared. After intratumoural injection, the LNs were taken out at 4 h, 8 h, and 12 h for NIRF imaging. For verifying the effect of different lipid materials in the cholesterol transport on the LNs accumulation capacity, the lipid material lecithin high potency and egg yolk lecithin were selected as representatives. After intratumoural injection, the LNs were taken out at 12 h for NIRF imaging.

Furthermore, we evaluated LNs accumulation of IL-15. Cy5.5-BSA was selected to imitate IL-15. Similar to the LNs accumulation of N-I-Lip, mice were intravenously injected with free Cy5.5-BSA, MMP2-sensitive liposomes and MMP2-non-sensitive liposomes. At 4 h, 8 h and 12 h, the mice were euthanized the acquire the LNs for in vitro imaging.

### In vitro cytotoxicity

Cytotoxicity in B16F10 cells was investigated by MTT assay. Briefly, $5 \times 10^3$ B16F10 cells (150 μL) per well were seeded in 96-well plates and incubated overnight. Subsequently, different concentrations of Blank-Lip, IR780, 1-MT, IL-15 and NIL-IM-Lip were added to the wells and further incubated for 48 h. The photothermal therapy group was irradiated with a laser (808 nm) at 1.0 W/cm$^2$ for 5 min. Then, 20 μL of MTT (5 mg/mL) was added to each well for another 4 h of incubation. Afterwards, the medium was removed and substituted with 200 μL of DMSO. Cell viability was measured with a microplate reader at 570 nm.

### Cell apoptosis and cellular uptake analysis

B16F10 cells (CT26 cells or MC38 cells) were seeded in 12-well plates at a density of $2 \times 10^5$/well and incubated overnight. Then, fresh medium containing PBS, IR780, IR780 + Laser (IR780 + L), IR780 + 1-MT + Laser (IR780 + 1-MT + L), IR780 + 1-MT + IL-15+Laser (IR780 + 1-MT + IL-15 + L) or NIL-IM-Lip+Laser (NIL-IM-Lip+L) was added. The photothermal therapy group was irradiated with a laser (808 nm) at 1.0 W/cm$^2$ for 5 min. The cells were collected and resuspended in PBS after incubation for 24 h. Apoptosis was analysed by flow cytometry after staining with PI (10 μL) and annexin V-FITC (5 μL).

Fluorescein isothiocyanate (FITC) was selected to label liposomes to cellular uptake analysis. HUVECs were seeded into 12-well plates and cultured overnight. The cells were then incubated with free FITC, FITC-Lip (F-Lip) or NGR-FITC-Lip (N-F-Lip) (27 μg/mL) for 1, 2 and 4 h. The cells were washed with cold PBS three times and fixed with 4% paraformaldehyde at 4 °C for 10 min. After washing with cold PBS three times, the cells were stained with DAPI for 15 min. Finally, images of the cells were captured by fluorescence microscopy. To quantify HUVEC uptake efficiency, the HUVECs were incubated with FITC-loaded liposomes (5 μg/mL) and evaluated by flow cytometry.

### Deep tumour penetration assay

The hanging drop method was used to prepare B16F10 3D tumour spheroids. Briefly, $1.5 \times 10^5$ B16F10 cells dispersed in 0.24% methylcellulose RPMI 1640 medium (20 μL) were transferred to the lids of 6-well plates. After one week of incubation, the B16F10 tumour spheroids were added to 12-well plates and cultured with FITC-Lip-Large (F-Lip-L) and FITC-Lip-Small (F-Lip-S) (FITC: 20 μg/mL). After incubation for 6 h,

the B16F10 tumour spheroids were washed with cold PBS. Finally, the deep tumour penetration ability was determined using an LSM 900 microscope.

## In vitro immunogenic cell death induction and DC maturation

B16F10 cells (CT26 cells or MC38 cells) were seeded in 12-well plates and cultured overnight. Then, the cells were incubated with PBS, IR780, IR780 + L, IR780 + 1-MT + L, IR780 + 1-MT + IL-15 + L or NIL-IM-Lip+L for 4 h. The IR780 + L, IR780 + 1-MT + L, IR780 + 1-MT + IL-15 + L and NIL-IM-Lip+L groups were irradiated with an 808 nm laser at 1.0 W/cm$^2$ for 5 min, followed by incubation for another 4 h to promote CRT exposure. Then, the cells were fixed with 4% paraformaldehyde for 10 min at 4 °C. Finally, the cells were incubated with primary antibodies and an AF594-conjugated secondary antibody (AF488-conjugated secondary antibody on CT26 and MC38 cells) for 30 min each and stained with DAPI. Images were captured with an inverted fluorescence microscope. The quantitative expression of CRT was determined by flow cytometry. For intracellular HMGB1 detection, B16F10 cells were seeded in 12-well plates and cultured overnight. After incubation for 4 h with different formulations, the groups with laser irradiation were treated with an 808 nm laser at 1.0 W/cm$^2$ for 5 min. Following further incubation for another 12 h, the cells were fixed with 4% paraformaldehyde and permeabilized with 0.1% Triton X-100 for 10 min. Finally, the cells were incubated with primary antibodies and an AF594-conjugated secondary antibody for 30 min each and stained with DAPI. Images were captured with an inverted fluorescence microscope. In addition, the HMGB1 released into the supernatant and ATP were collected and analysed with an ELISA kit (Solarbio, China) and an ATP assay kit. For ROS evaluation, a total of $1 \times 10^5$ B16F10 cells per well were seeded in 12-well plates and cultured overnight. Subsequently, PBS, IR780, IR780 + L, IR780 + 1-MT + L, IR780 + 1-MT + IL-15 + L or NIL-IM-Lip+L were added for another 4 h of incubation. Following incubation with DCF (fluorescent probe for ROS) for 1 h, the IR780 + L, IR780 + 1-MT + L, IR780 + 1-MT + IL-15 + L and NIL-IM-Lip+L groups were irradiated with an 808 nm laser at 1.0 W/cm$^2$ for 5 min. Finally, an inverted fluorescence microscope was selected to image the cells. ROS production was quantitatively evaluated by flow cytometry.

Then, bone marrow-derived dendritic cells (BMDCs) were extracted from the bones of female C57BL/6 mice (6–8 week). Then, the obtained BMDCs were cultured in 6-well plates with RPMI 1640 medium containing GM-CSF (20 μg/mL) and IL-4 (20 μg/mL). Then, immature DCs were seeded in 12-well plates, and B16F10 cells were seeded in a transwell chamber. B16F10 cells were treated with different formulations and then cocultured with the DCs using a transwell system. Mature DCs were stained with an FITC anti-mouse CD11c antibody, PE anti-mouse CD80 antibody and APC anti-mouse CD86 antibody for flow cytometry analysis. In addition, the collected culture medium was centrifuged to analyse the amount of TNF-α secreted.

## In vitro NK killing ability

Briefly, NK cells were obtained from the spleens of female C57BL/6 mice (6–8 week) using the MojoSort™ Mouse NK Cell Isolation Kit under aseptic conditions. Subsequently, the purity of the obtained NK cells was determined by flow cytometry. The in vitro NK killing ability was measured by MTT assay. Briefly, NK cells were cultured with IL-2, IL-15 and NIL-IM-Lip after preincubation with the MMP2 enzyme. After 3 days, NK cells were added to B16F10 cells cultured in 96-well plates (5000 cells per well) overnight at ratios of 10:1 and 20:1. Following 6 h of incubation, MTT solution (5 mg/mL, 20 μL) was added to each well, and the cells were incubated for another 4 h. Finally, 200 μL of DMSO was added to each well, and the absorbance of each well was measured with a microplate reader (Model 680; Bio–Rad, CA, USA).

## In vivo antitumour postsurgical recurrence efficacy

B16F10 tumour-bearing female C57BL/6 mice were used to evaluate the antitumour postsurgical recurrence efficacy. A total of $1 \times 10^6$ B16F10 cells were subcutaneously injected into the right axilla of the mice to establish the model. After 10 days, 90% of the tumour was surgically removed, and the wounds were sutured. The mice were randomly divided into 14 groups (n = 5 biologically independent animals per group): (1) NS, (2) Blank-Lip, (3) IR780, (4) IR780 + L, (5) 1-MT, (6) IL-15, (7) N-I-Lip+L, (8) IR780 + 1-MT + L, (9) N-IM-Lip+L, (10) IR780 + IL-15 + L, (11) NIL-I-Lip+L, (12) IR780 + 1-MT + IL-15 + L, (13) IL-IM-Lip+L, and (14) NIL-IM-Lip+L. Moreover, we evaluated the efficacy of NIL-IM-Lip without laser irradiation. The NS and NIL-IM-Lip+L groups were selected as control groups. The dosages of IR780, 1-MT and IL-15 were 1 mg/kg, 2.7 mg/kg and 1 μg/per mouse, respectively. The mice were treated every 4 days, and the tumour volumes and body weights were measured every 2 days. The mice in the photothermal therapy groups were irradiated with a laser (808 nm) at 1.0 W/cm$^2$ for 5 min after 12 h administration of the different formulations. In addition, the mice were imaged at -1, 0, 7 and 14 days. The spleens were taken out to evaluate the memory T cells, which were marked using an APC anti-mouse CD3 antibody, PE anti-mouse CD8a antibody, FITC anti-mouse CD62L antibody and PerCP/Cyanine 5.5 anti-mouse CD44 antibody. The tumour volume (V) was calculated as:

$$V = W^2(\text{width}) \times L(\text{length})/2 \qquad (3)$$

## In vivo antitumour activity

B16F10 tumour-bearing female C57BL/6 mice were used to evaluate antitumour activity. B16F10 cells ($8 \times 10^5$) were subcutaneously injected into the right axilla of the mice. After 10 days, the mice were randomly divided into 14 groups (n = 6 biologically independent animals per group): (1) NS, (2) Blank-Lip, (3) IR780, (4) IR780 + L, (5) 1-MT, (6) IL-15, (7) N-I-Lip+L, (8) IR780 + 1-MT + L, (9) N-IM-Lip+L, (10) IR780 + IL-15 + L, (11) NIL-I-Lip+L, (12) IR780 + 1-MT + IL-15 + L, (13) IL-IM-Lip+L, and (14) NIL-IM-Lip+L. The dosages of IR780, 1-MT and IL-15 were 1 mg/kg, 2.7 mg/kg and 1 μg/mouse, respectively. The mice in the photothermal therapy groups were irradiated with a laser (808 nm) at 1.0 W/cm$^2$ for 5 min after 12 h administration of the different formulations. The mice used for survival period studies were treated in the same manner. The mice were treated every 4 days. Tumour volumes and body weights were measured every 2 days. On Day 16, the mice were euthanized, and the tumours were excised and photographed. Similar to the antitumour postsurgical recurrence efficacy assay, NIL-IM-Lip was also evaluated.

Furthermore, the efficacy of NIL-IM-Lip+L combined with a PD-1 mAb was evaluated in the B16F10 model and CT26 model. The B16F10 model was constructed by subcutaneously injecting $8 \times 10^5$ B16F10 cells into the right axilla of female C57BL/6 mice. The CT26 model was constructed by subcutaneously injecting $1 \times 10^6$ CT26 cells into the right axilla of female BALB/c mice. The mice were divided into 3 groups (n = 5 biologically independent animals per group): (1) NS, (2) PD-1, (3) NIL-IM-Lip+L, and (4) NIL-IM-Lip+PD-1 + L. The treatment procedure was similar to that describe before. The PD-1 mAb was intraperitoneally injected at a dose of 5 mg/kg.

## In vivo immunization study

After evaluating the in vivo antitumour activity, the tumours and LNs were excised and filtered through a copper network. Lymphocytes were separated from the tumour mixture with a 40% Percoll solution. CD3$^+$CD4$^+$ and CD3$^+$CD8$^+$ T cells were stained with an APC anti-mouse CD3 antibody, FITC anti-mouse CD4 antibody and PE anti-mouse CD8a antibody. DCs were stained with an PE anti-mouse CD11c antibody, FITC anti-mouse CD80 antibody and PerCP/Cyanine5.5 anti-mouse CD86 antibody. NK cells were stained with an APC anti-mouse CD3

antibody and PE anti-mouse CD49b antibody. Treg cells were stained with an PE anti-mouse CD4 antibody and AF647 anti-mouse FOXP3 antibody. CTLs were stained with an PE anti-mouse CD8 antibody and APC anti-mouse IFN-γ antibody. The changes in the immune cells were measured by flow cytometry. In addition, the levels of cytokines in the tumours, including IFN-γ, TGF-β, IL-10, IL-12 and IL-6, were examined with ELISA kits. Furthermore, the excised tumours were cut and filtered through a copper network to obtain a homogenate. Then, the homogenate was incubated with trichloroacetic acid (30%) for 30 min at 50 °C. The concentrations of tryptophan (Trp) and kynurenine (Kyn) were measured by HPLC at 360 and 280 nm after centrifugation. Moreover, the gating strategies for immune cells were provided in Supplementary Figs. 35-37.

## Immunohistochemical analysis
Tumour tissues and major organs were fixed in 4% formaldehyde and embedded in paraffin. The tumour sections were stained with haematoxylin and eosin (H&E), Ki67, and TUNEL. The major organs were stained with H&E. Furthermore, IDO1 was observed in the tumour sections and LNs. In addition, CRT and HMGB1 in the tumour sections were detected.

## Haemolysis assay
NIL-IM-Lip was incubated with a red blood cell (RBC) (2%, v/v) suspension at $37 \pm 0.5\,°C$ for 3 h. The NS group and distilled water group were used as the negative control and positive control, respectively. The absorbance of haemoglobin was measured with a UV–vis spectrophotometer at 576 nm.

## Statistical analysis
All data were presented as the mean ± SD. The statistically significant differences were analyzed by the Student's t-test (two groups) and a one-way analysis of variance (ANOVA) (multiple groups). The $P$ value was calculated with the help of GraphPad Prism 8.0.1 and Microsoft Excel 2019 software. $*P < 0.05$, $**P < 0.01$, $***P < 0.001$, $\#P < 0.001$ were considered statistically significant.

## Reporting summary
Further information on research design is available in the Nature Portfolio Reporting Summary linked to this article.

## Data availability
The all raw data generated in this study are provided in the Supplementary Information /Source Data file. Source data are provided with this paper.

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

## Acknowledgements

Our work was supported by National Natural Science Foundation of China (No. 82173757 to N.Z., No. 82173756 to Y.L.) and the Young Scholar Program of Shandong University (YSPSDU, 2017WLJH40 to Y.L.). We are grateful for Translational Medicine Core Facility of Shandong University and Pharmaceutical biology sharing platform of Shandong University for supporting this work.

## Author contributions

S.F., N.Z., and Y.L. conceived the research and designed the experiments. S.F. performed the experiments and analyzed the data. X.S. contributed to the preparation of nanoinducer and photothermal performance. L.C, S.L., H.Y., HZ.Y., and Q.M. contributed to the NIRF imaging and antitumour treatment. T.G. contributed to the analysis of immune cells. W.M., S.L, and Z.Z. contributed to the design of the nanoinducer. X.L. contributed to the synthesis of functional materials and MMP2 sensitive enzymatic digestion. S.F., N.Z. and Y.L. wrote the paper. All authors analyzed and discussed the data, and the final draft was approved by all authors.

## Competing interests

The authors declare no competing interests.
