## [Peer Review File · Nature Communications]

Temperature sensitive liposome based cancer nanomedicine enables tumour lymph node immune microenvironment remodellingEditorial Note: This manuscript has been previously reviewed at another journal that is not operating a transparent peer review scheme. This document only contains reviewer comments and rebuttal letters for versions considered at *Nature Communications*.

REVIEWER COMMENTS

Reviewer #1 (Remarks to the Author):

They did not address most of the comments from the previous reviews. Their statistical analyses were based on the Students' T-test which is wrong. Besides, they did not include key control groups that were suggested in most studies. Moreover, their system is too complicated. CT26 tumors are not considered cold tumors in most papers. I recommend they submit this to another subspecialty journal.

Reviewer #3 (Remarks to the Author):

Fu et al. developed a temperature sensitive immunomodulatory inducer to remodel the tumor-lymph node immune microenvironment. They suggest an innovative immunotherapy approach and combine with anti PD-1 treatment. Although their results are striking, there are some points need to be addressed:

MAJOR:

1. In figure 4a, fluorescence spillovers were not corrected appropriately. Therefore, compensation of the Annexin V and PI should be rechecked and corrected.
2. The authors used mouse cell lines (B16F10 and CT26) for the in vitro and in vivo experiments, but they performed uptake assays on HUVEC cells. Instead, it would be better to use mouse endothelial cells.
3. In figure 4d, they gave the percentages of HUVEC cells that engulfed the formulations. Although they obtained statistically significant difference, in figure 4e, representative flow cytometry histograms seem quite similar for 4 different conditions. Moreover, after 4 hours of incubation, the percentages of cellular uptake are approximately 60% but in the flow cytometry histograms the fluorescent intensity seem very high (around at 104). To clarify this, they need to add the flow cytometry histogram of unstained cells as well.
4. In figure 4f, especially for the images at 4h, the signals of NF-Lip seem very intense and there is a high background fluorescence. Since, there is no drastic difference observed in flow cytometric uptake assay, this dramatically intense fluorescent signals for NF-Lip makes these images questionable. Did they use the same exposure time for each condition while capturing the images?
5. In figure 6g, in addition to the representative images of Ki67 and TUNEL staining, the positivity of Ki67 and TUNEL would be quantified in mm² of the tissues to strengthen the authors' claim.
6. In figure 8, the combination therapy of NIL-IM-Lip+L with anti-PD-1 mAb seems superior to NIL-IM-Lip+L only. However, they did not perform experiments with anti-PD-1 mAb only. If they would indicate the superiority of their combination therapy over anti-PD-1 therapy, it would improve the manuscript.

MINOR:

1. In the study, there are lots of different types of formulations. Although abbreviations were given in the text, they should also be given in the figure legends to make it clearer for the readers.

Response to reviewers' comments:

Reviewer #1 (Remarks to the Author):

Comments:

1. Their statistical analyses were based on the Students' T-test which is wrong.

Response: Thanks for your comments. The statistical significance of the effects in the manuscript have been revised. For the comparison among three or more groups, one-way ANOVA analysis of variance with Tukey's post hoc test were selected. We have checked the whole manuscript and revised the experimental statistical analysis methods in the Figure legend, which had no effect on the main results and conclusions. The revised parts were marked in red.

2. Besides, they did not include key control groups that were suggested in most studies.

Response: Thanks for your comments. For the *in vitro* experiments, the key control group of the IR780+1-MT+IL-15+Laser was added in the revised manuscript. The *in vitro* evaluations in our study included: cell apoptosis assay, ICD evaluation (CRT exposure, HMGB1 release, ATP secretion), BMDC maturation assay and ROS evaluation. So, we set up 6 groups, including the PBS, IR780, IR780+Laser, IR780+1-MT+Laser, IR780+1-MT+IL-15+Laser and NIL-IM-Lip+Laser groups follow your suggestions. The results of the revised relevant figures were listed as below and were supplemented in the revised manuscript. The results of the cell apoptosis assay on B16F10 cells, CT26 cells and MC38 cells were shown in **Fig. 4a, Supplementary Fig. 30 and 31**. The results of CRT exposure, HMGB1 release, ATP secretion on B16F10 cells were shown in **Fig.5a,d, Fig.5c,e, Fig.5f**. The results of BMDC maturation rate and the TNF- α secretion were shown in **Fig.5h,l** and **Fig.5k**. The results of CRT exposure on CT26 and MC38 cells were shown in **Supplementary Fig. 32 and 33**. The results of ROS generation was shown in **Supplementary Fig. 15**. The results in the **Fig. 4a, Supplementary Fig.30 and 31** showed that the NIL-IM-Lip+L group had higher cell apoptosis rate than the IR780+1-MT+IL-15+L group on B16F10 cells, CT26 cells, MC38 cells. Then, as shown in **Fig.5a,d**, the more red fluorescence and the higher quantitative fluorescent signal in the NIL-IM-Lip+L group demonstrated that the NIL-IM-Lip+L could promote more CRT exposure than that in the IR780+1-MT+L and IR780+1-MT+IL-15+L groups ($P < 0.001$). The fewer HMGB1 nuclei (**Fig. 5c**), more HMGB1 release (**Fig. 5e**) ($P < 0.01$) and the more ATP secretion (**Fig. 5f**) ($P < 0.05$) also confirmed the enhanced ICD effect in the NIL-IM-Lip+L group. Then, the BMDC maturation rate and the TNF- α secretion were evaluated. As shown in **Fig. 5h,l**, the NIL-IM-Lip+L group displayed the greatest maturation of DCs ($P < 0.01$). Besides, the NIL-IM-Lip+L group secreted a higher level of TNF- α into the culture supernatant than the IR780+1-MT+L group and IR780+1-MT+IL-15+L group ($P < 0.05$) (**Fig. 5k**). The CRT exposure were also evaluated on the CT26 cells and MC38 cells. As shown in the

Supplementary Fig.32 and 33, the NIL-IM-Lip+L group could promote the CRT exposure on both CT26 and MC38 cells. The results of ROS generation were also detected and shown in **Supplementary Fig. 15**. The groups with irradiation of laser showed significantly increased fluorescence compared with the groups without laser irradiation. The NIL-IM-Lip+L group showed higher fluorescence than the IR780+1-MT+L group and IR780+1-MT+IL-15+L group ($P < 0.001$).

Fig. 4a. Flow cytometric apoptosis analysis of B16F10 cells after treatment with PBS, IR780, IR780+L, IR780+1-MT+L, IR780+1-MT+IL-15+L and NIL-IM-Lip+L. The IR780+L, IR780+1-MT+L, IR780+1-MT+IL-15+L and NIL-IM-Lip+L groups were irradiated with an 808 nm laser for 5 min (1.0 W/cm²).

Supplementary Fig. 30. Flow cytometric apoptosis analysis of CT26 cells after

treatment with PBS, IR780, IR780+L, IR780+1-MT+L, IR780+1-MT+IL-15+L and NIL-IM-Lip+L. The IR780+L, IR780+1-MT+L, IR780+1-MT+IL-15+L and NIL-IM-Lip+L groups were irradiated with an 808 nm laser for 5 min (1.0 W/cm²).

Supplementary Fig. 31. Flow cytometric apoptosis analysis of MC38 cells after treatment with PBS, IR780, IR780+L, IR780+1-MT+L, IR780+1-MT+IL-15+L and NIL-IM-Lip+L. The IR780+L, IR780+1-MT+L, IR780+1-MT+IL-15+L and NIL-IM-Lip+L groups were irradiated with an 808 nm laser for 5 min (1.0 W/cm²).

Fig. 5. The properties of NIL-IM-Lip+L on the ICD effect, DC maturation and the killing effects of NK cells *in vitro*. **a**, Fluorescence microscopy images of CRT exposure to B16F10 cells. The IR780+L, IR780+1-MT+L, IR780+1-MT+IL-15+L and NIL-IM-Lip+L groups were irradiated with an 808 nm laser for 5 min (1.0 W/cm^2). **b**, Schematic illustration of the ICD effect to induce tumour cell death, including CRT exposure, HMGB1 release, and ATP secretion, which could promote the maturation of DCs. **c**, Fluorescence microscopy images of HMGB1 release from B16F10 cells. The IR780+L, IR780+1-MT+L, IR780+1-MT+IL-15+L and NIL-IM-Lip+L groups were irradiated with an 808 nm laser for 5 min (1.0 W/cm^2). **d**, Flow cytometric analysis of CRT exposure to B16F10 cells ($n = 3$ biologically independent experiments). **e**, Quantitative ELISA analysis of HMGB1 release into the B16F10 culture medium supernatant ($n = 3$ biologically independent experiments). **f**, Quantitative ELISA analysis of ATP secretion into the B16F10 culture medium supernatant ($n = 3$ biologically independent experiments). **g**, Schematic illustration of the transwell system used for DC maturation detection. **h, i**, Flow cytometric analysis of the ratio of mature DCs after coculture with B16F10 cells prestimulated with different formulations ($n = 3$ biologically independent

experiments). **i**, Schematic illustration of the experimental process to determine the killing ability of NK cells. **j**, B16F10 cells killed by NK cells after pretreatment with different formulations (n = 3 biologically independent experiments). **k**, Quantitative ELISA analysis of TNF- α secretion in the DC culture medium supernatant (n = 3 biologically independent experiments). Statistical significance was calculated by one-way ANOVA analysis of variance with Tukey's post hoc test. * $P < 0.05$, ** $P < 0.01$, *** $P < 0.001$, ns $P > 0.05$.

Supplementary Fig. 32. Fluorescence microscopy images of CRT exposure to CT26 cells. The IR780+L, IR780+1-MT+L, IR780+1-MT+IL-15+L and NIL-IM-Lip+L groups were irradiated with an 808 nm laser for 5 min (1.0 W/cm^2). Scale bar: 100 μm .

Supplementary Fig. 33. Fluorescence microscopy images of CRT exposure to MC38 cells. The IR780+L, IR780+1-MT+L, IR780+1-MT+IL-15+L and NIL-IM-Lip+L groups were irradiated with an 808 nm laser for 5 min (1.0 W/cm^2). Scale bar: 100 μm .

Supplementary Fig. 15. NIL-IM-Lip+L induced stronger ROS generation *in vitro*. **a**, The fluorescence microscopy images of ROS generation in B16F10 cells. Scale bar: 100 μm. **b**, The flow cytometric analysis of ROS generation in B16F10 cells (n = 3 biologically independent experiments). Statistical significance was calculated by one-way ANOVA analysis of variance with Tukey's post hoc test. *** $P < 0.001$

3. Moreover, their system is too complicated.

Response: Thanks for your comments. The combination therapy or the concomitant drugs has been widely used in the clinic today^{1,2}. The latest cancer treatment guidelines published by National Comprehensive Cancer Network (NCCN) mentioned that the multi-drug combination, such as chemoimmunotherapy, the combinational therapy of chemical drug, the combinational therapy of immune drug, has been widely used as clinical first-line treatment, which indicated that the combination therapy was one of the major strategies for the treatment of cancer. Considering the significant differences in the pharmacokinetic process of different drugs *in vivo*, sequential administration greatly reduced the antitumour effects, which lead a higher dose of drugs to guarantee the effects^{3,4}. Unfortunately, the higher dose of drugs accompanied with inevitable accumulated side effects⁵. Pharmacy was developed to serve the clinic by constructing intelligent drug delivery vehicles, which will make the clinical combination treatments easier and improve the patient compliance. Multi-drug co-delivery system has become one of the focus of drug discovery.

We very agree with your concern about the complexity of the delivery system, so we selected the materials for clinical use and the preparation process for clinical preparations to prepare the NIL-IM-Lip.

In our design, the functions were achieved by easy-to-implement functional materials. Because DSPE-PEG, DSPE and PEG are all safe materials and approved by FDA for clinical usage, both DSPE-Hyd-PEG₅₀₀₀-NGR and DSPE-PEG₂₀₀₀-MMP2 pep-IL-15 was acceptable functional materials. And the synthesis process was easy to implement, our system was not complicated to achieve pH/MMP2 sensitive functions. Otherwise, the temperature-sensitive function was obtained by simply using the

temperature-sensitive lipid materials DPPC.

Besides, the preparation process of NIL-IM-Lip was simple and the market was huge. We selected liposome, which was widely used in clinic, as the carrier to achieve co-delivery of drugs⁶. The global sales of liposome products reached \$5.152 billion in 2017, which indicated the huge market of liposome products. CPX-351(Vyxeos), the world's first co-loaded liposome, was approved in the United States for the treatment of acute myeloid leukemia (AML) in 2017. CPX-351 was the co-loaded liposome with a 5:1 fixed drug ratio of cytarabine and daunorubicin. Among the CPX-351, cytarabine was a hydrophilic drug and daunorubicin was a hydrophobic drug. Hence, the liposome co-loading IR780 (hydrophobic drug) and 1-MT (hydrophilic drug) in our study had the potential for clinical translation. Moreover, the preparation process of NIL-IM-Lip has not changed compared with the liposome products (AmBisome, Visudyne and Shingrix). The modification of the DSPE-PEG also used the conventional method, which would not enhance the complexity of the preparation process.

In summary, our system used the easy-to-implement functional materials and simple preparation process to prepare the NIL-IM-Lip, which could achieve multiple functions.

Reference

1. Meric-Bernstam F, Larkin J, Tabernero J, Bonini C. Enhancing anti-tumour efficacy with immunotherapy combinations. *Lancet*. 2021;397(10278):1010-1022.
2. Bayat Mokhtari R, Homayouni TS, Baluch N, et al. Combination therapy in combating cancer. *Oncotarget*. 2017;8(23):38022-38043.
3. Poon W, Kingston BR, Ouyang B, Ngo W, Chan WCW. A framework for designing delivery systems. *Nat Nanotechnol*. 2020;15(10):819-829.
4. Shi J, Kantoff PW, Wooster R, Farokhzad OC. Cancer nanomedicine: progress, challenges and opportunities. *Nat Rev Cancer*. 2017;17(1):20-37.
5. Senapati S, Mahanta AK, Kumar S, Maiti P. Controlled drug delivery vehicles for cancer treatment and their performance. *Signal Transduct Target Ther*. 2018;3:7.
6. Filipczak N, Pan J, Yalamarty SSK, Torchilin VP. Recent advancements in liposome technology. *Adv Drug Deliv Rev*. 2020;156:4-22.

4.CT26 tumors are not considered cold tumors in most papers.

Response: Thanks for your comments. The “cold” tumours have a low immune response with less infiltrated immune cells and more immunosuppressive cells, such as Treg cells. The “hot” tumours are infiltrated with more activating immune cells (such as CD8⁺ T cells) and have a better response to immunotherapy. The review published in the **Nature reviews gastroenterology & hepatology** (10.1038/s41575-019-0126-x) described “Colorectal cancer (CRC) can be categorized into tumours that are mismatch-

repair-deficient or have high levels of microsatellite instability (dMMR–MSI-H; ~15%) and mismatch-repair-proficient or microsatellite instability-low tumours (pMMR–MSI-L; ~85%)”. Only 15% dMMR–MSI-H colorectal cancer patients responded to PD-1 antibody, whereas other pMMR–MSI-L colorectal cancer patients had a very low response rate to PD-1 antibody. Most of the colorectal cancer patients can’t benefit from the immunotherapy alone.

A research named “Ginseng-derived nanoparticles potentiate immune checkpoint antibody efficacy by reprogramming the cold tumour microenvironment” published in the **Molecular Therapy** (10.1016/j.ymthe.2021.08.028) described “PD-1 mAb alone significantly inhibited the growth of hot tumours such as B16-F10 murine melanoma (Figures S1A and S1B), but mice bearing CT26 murine colon tumour and 4T1 murine breast tumour were resistant to PD-1 mAb treatment”, which defined the B16F10 model as hot tumour and the CT26 model as cold tumour. In addition, another research published in the **Acta Pharm Sin B** (10.1016/j.apsb.2021.09.022) mentioned “Generally, CRC can be considered as a “cold” tumour and the majority of CRC patients are poorly immune responsive; merely 15% of CRCs are dMMR–MSI-H (mismatch-repair-deficient and microsatellite instability-high), which is a positive prognosis biomarker for immunotherapy” in the Introduction and the CT26 model was selected to perform the experiments.

We also evaluated the results of the CTL/Treg ratio, CD8⁺T/Treg ratio on the both B16F10 model and CT26 model for the comparison of immune status in different tumour models. The higher CTL/Treg ratio and CD8⁺T/Treg ratio represented more activating immune cells and less immunosuppressive cells infiltration in the tumour microenvironment, and the tumour model was defined as “hot” tumour. The low CTL/Treg ratio and CD8⁺T/Treg ratio represented less activating immune cells and more immunosuppressive cells infiltration in the tumour microenvironment, and the tumour model was defined as “cold” tumour. As shown in the figure below, the higher ratio of CTL/Treg and CD8⁺T/Treg in the B16F10 model compared than CT26 model ($P < 0.05$, $P < 0.001$). Hence, the CT26 model was considered as cold tumour model.

The ratio of CTL/Treg and CD8⁺T/Treg in the tumour (NS group) both in the B16F10

and CT26 model (n = 3 biologically independent experiments). * $P < 0.05$, *** $P < 0.001$.

Reviewer #3 (Remarks to the Author):

Comments

Fu et al. developed a temperature sensitive immunomodulatory inducer to remodel the tumor-lymph node immune microenvironment. They suggest an innovative immunotherapy approach and combine with anti PD-1 treatment. Although their results are striking, there are some points need to be addressed:

MAJOR:

1. In figure 4a, fluorescence spillovers were not corrected appropriately. Therefore, compensation of the Annexin V and PI should be rechecked and corrected.

Response: Thanks for your comments. The fluorescence spillovers of the cell apoptosis assay have been corrected. We have rechecked and corrected the compensation of the Annexin V and PI and the **Fig. 4a**, and the result was supplemented in the revised manuscript.

Fig. 4a. Flow cytometric apoptosis analysis of B16F10 cells after treatment with PBS, IR780, IR780+L, IR780+1-MT+L, IR780+1-MT+IL-15+L and NIL-IM-Lip+L. The IR780+L, IR780+1-MT+L, IR780+1-MT+IL-15+L and NIL-IM-Lip+L groups were irradiated with an 808 nm laser for 5 min (1.0 W/cm²).

2. The authors used mouse cell lines (B16F10 and CT26) for the *in vitro* and *in vivo* experiments, but they performed uptake assays on HUVEC cells. Instead, it would be better to use mouse endothelial cells.

Response: Thanks for your comments and we very agree with your suggestions.

Indeed, the uptake assay to use the mouse endothelial cells would be better than HUVECs. CD13, a Zn²⁺ dependent metalloprotease, was highly expressed in many tumour and angiogenic cells but not activated in normal endothelial cells^{1,2}. NGR could target the CD13 receptor in tumour neovascularization for effective tumour accumulation^{3,4}. We checked many literatures and researches and no mouse CD13 high-expression vascular endothelial cell line was found. The normal mouse endothelial cells was not highly expressed the CD13 receptor. Hence, we have to consider another endothelial cell with high CD13 expression. Human umbilical vein endothelial cells (HUVEC), a type of human vascular endothelial cell with high CD13 receptor expression, were often used as model cells to test the targeting of NGR⁵⁻⁷. For example, Angelo Corti et. al used the HUVECs to evaluated the binding with the NGR peptide in *Advanced Functional Materials*⁸. Ting Kang and his group used the HUVECs to evaluated the cellular uptake of iNGR-modified PEG-PLGA nanoparticles in *Biomaterials*⁹. Besides, the tumour cells with high CD13 expression were also used to evaluated the NGR-mediated tumour vasculature accumulation. For example, Guowei Wang et.al used the Biopsy xenograft of Pancreatic Carcinoma line-3 (BxPC3) to evaluated the cellular uptake of the NGR modified LPGL in *Advanced Matirials*¹⁰. Finally, considering to get closer to the tumor neovascularization endothelial cells, HUVEC was selected for cellular uptake assay which could prove the CD13-targeting function of NGR modification.

Reference

1. Curnis F, Sacchi A, Borgna L, Magni F, Gasparri A, Corti A. Enhancement of tumor necrosis factor alpha antitumor immunotherapeutic properties by targeted delivery to aminopeptidase N (CD13). *Nat Biotechnol.* 2000;18(11):1185-1190.
2. Bhagwat SV, Lahdenranta J, Giordano R, Arap W, Pasqualini R, Shapiro LH. CD13/APN is activated by angiogenic signals and is essential for capillary tube formation. *Blood.* 2001;97(3):652-659.
3. Rong L, Zhang Y, Li WS, Su Z, Fadhil JI, Zhang C. Iron chelated melanin-like nanoparticles for tumor-associated macrophage repolarization and cancer therapy. *Biomaterials.* 2019;225:119515.
4. Seidi K, Jahanban-Esfahlan R, Monhemi H, et al. NGR (Asn-Gly-Arg)-targeted delivery of coagulase to tumor vasculature arrests cancer cell growth. *Oncogene.* 2018;37(29):3967-3980.
5. Yang R, Zhang Z, Fu S, et al. Charge and Size Dual Switchable Nanocage for Novel Triple-Interlocked Combination Therapy Pattern. *Adv Sci (Weinh).* 2020;7(18):2000906.
6. Guan L, Zhang Z, Gao T, et al. Depleting Tumor Infiltrating B Cells to Boost Antitumor Immunity with Tumor Immune-Microenvironment Reshaped Hybrid Nanocage. *ACS Nano.* 2022;16(3):4263-4277.
7. An S, Jiang X, Shi J, et al. Single-component self-assembled RNAi nanoparticles

functionalized with tumor-targeting iNGR delivering abundant siRNA for efficient glioma therapy. *Biomaterials*. 2015;53:330-340.

8. Corti A, Gasparri AM, Ghitti M, et al. Glycine N-methylation in NGR-Tagged Nanocarriers Prevents Isoaspartate formation and Integrin Binding without Impairing CD13 Recognition and Tumor Homing. *Adv Funct Mater*. 2017;27(36):1701245.

9. Kang T, Gao X, Hu Q, et al. iNGR-modified PEG-PLGA nanoparticles that recognize tumor vasculature and penetrate gliomas. *Biomaterials*. 2014;35(14):4319-4332.

10. Wang G, Jiang Y, Xu J, et al. Unraveling the Plasma Protein Corona by Ultrasonic Cavitation Augments Active-Transporting of Liposome in Solid Tumor. *Adv Mater*. 2022;e2207271.

3. In figure 4d, they gave the percentages of HUVEC cells that engulfed the formulations. Although they obtained statistically significant difference, in figure 4e, representative flow cytometry histograms seem quite similar for 4 different conditions. Moreover, after 4 hours of incubation, the percentages of cellular uptake are approximately 60% but in the flow cytometry histograms the fluorescent intensity seem very high (around at 10^4). To clarify this, they need to add the flow cytometry histogram of unstained cells as well.

Response: Thanks for your comments. We have rechecked and corrected the mistake in the **Fig. 4e**. In the previous manuscript, from top to bottom were the 4. NF-Lip, 3. NF-Lip with preincubation, 2. F-Lip, 1. NS groups, which was reversed with the labeling order. Besides, we also supplemented the histogram of unstained cells. The rechecked **Fig. 4e** were supplemented in the revised manuscript. In order to better show the difference of **Fig. 4e**, we also provided the raw data image of the flow cytometry histogram.

The revised **Fig. 4e** were shown as below:

Fig. 4d-e, Flow cytometric analysis of HUVEC uptake. FITC was selected to replace 1-MT for the cellular uptake assay (n = 3 biologically independent experiments).

The raw data image of the flow cytometry histogram was shown as below:

4. In figure 4f, especially for the images at 4h, the signals of NF-Lip seem very intense and there is a high background fluorescence. Since, there is no drastic difference observed in flow cytometric uptake assay, this dramatically intense fluorescent signals for NF-Lip makes these images questionable. Did they use the same exposure time for each condition while capturing the images?

Response: Thanks for your comments. The cellular uptake assay was keeping the same exposure time when capturing the images. In order to enhance the display effect of the images in the **Fig. 4f**, we uniformly enhanced the brightness of the images in **Fig. 4f**. The high background fluorescence in the NF-Lip at 4 h (**Fig. 4f**) was because the uniformly over-enhanced brightness of images, which resulted in significant

enhancement of the background fluorescence of the NF-Lip group at 4 h. We have uniformly revised the brightness of the images in **Fig. 4f**, and the revised **Fig.4f** were shown in the revised manuscript.

In this study, the “cellular uptake” experiments were carried out in three times (only one of the pictures was shown in the manuscript). The images of NF-Lip group at 4 h in HUVEC cells for three times experiments were shown as below figure. The results of the flow cytometric uptake assay were the quantitative data of whole cells. The image was a magnified field of camera vision, which displayed part of a complete field of vision. In the magnified field of vision, the images may display fewer cells. Moreover, the fluorescence signal in the 4 h NF-Lip appears particularly pronounced due to the excessive uniform enhancement of the image brightness. Therefore, the discrepancy between the fluorescent intensity of NF-Lip in Figure 4F and the results of the flow cytometric uptake assay due to the difference of the camera vision and the over-enhanced brightness. We corrected the high background fluorescence in the revised manuscript to reduce the discrepancy in vision between the fluorescence intensities and the flow cytometric uptake assay.

Fig. 4f. Fluorescence microscopy image of HUVEC uptake. Scale bar=200 μ m.

The images of NF-Lip group at 4 h in HUVEC cells for three times experiments.

5. In figure 6g, in addition to the representative images of Ki67 and TUNEL staining, the positivity of Ki67 and TUNEL would be quantified in mm² of the tissues to strengthen the authors' claim.

Response: Thanks for your comments. Indeed, the quantitative results of Ki67 and TUNEL staining images were more confirmable than the images. So, the quantitative results of the Ki67 and TUNEL staining images were supplemented as your suggestions. As shown in **Supplementary Fig. 20**, the NIL-IM-Lip+L group showed less positivity of Ki67 and more positive of TUNEL.

Supplementary Fig. 20. Quantification of (a) Ki67 and (b) TUNEL staining assays of tumour tissues. Statistical significance was calculated by one-way ANOVA analysis of

variance with Tukey's post hoc test. * $P < 0.05$, ** $P < 0.01$, # $P < 0.001$

6. In figure 8, the combination therapy of NIL-IM-Lip+L with anti-PD-1 mAb seems superior to NIL-IM-Lip+L only. However, they did not perform experiments with anti-PD-1 mAb only. If they would indicate the superiority of their combination therapy over anti-PD-1 therapy, it would improve the manuscript.

Response: Thanks for your comments. Anti-PD-1 therapy alone as the control was necessary for confirming the great antitumour effects of the combination therapy. The antitumour effects of the combination therapy containing the NS, PD-1, NIL-IM-Lip+L, NIL-IM-Lip+PD-1+L groups were supplemented follow your suggestions. The schedule of the *in vivo* administration approach in the B16F10 model and CT26 model were shown in **Fig. 8a** and **Fig. 8j**.

In the B16F10 model, smaller tumour volumes and lighter tumour weight were observed after NIL-IM-Lip+PD-1+L treatment than those in the NIL-IM-Lip+L group ($P < 0.001$) and PD-1 group ($P < 0.001$) (**Fig. 8b-e**). As shown in **Fig. 8e**, one mouse of NIL-IM-Lip+PD-1+L was cured and the NIL-IM-Lip+PD-1+L gave a 20% complete response (CR), which indicated the great potential of NIL-IM-Lip+L combined with the PD-1 mAb. Subsequently, the tumour and the lymph nodes were used to the analysis of immune cells. In the tumour tissues (**Fig. 8f,h**), the frequency of positive immune responders in the NIL-IM-Lip+PD-1+L group, including CD4⁺ T cells, CD8⁺ T cells, CTLs and NK cells, were higher than those in the NIL-IM-Lip+L group ($P < 0.001$, $P < 0.001$, $P < 0.05$, $P < 0.001$) and PD-1 group ($P < 0.001$, $P < 0.001$, $P < 0.001$, $P < 0.001$). The NIL-IM-Lip+PD-1+L group greatly suppressed negative Treg cells compared with NIL-IM-Lip+L group and PD-1 group ($P < 0.001$) (**Fig. 8f**). Consistent with changes in the immune state in the tumour tissues, NIL-IM-Lip+PD-1+L activated more CD4⁺ T cells, CD8⁺ T cells and NK cells and inhibited more Treg cells in the LNs (**Fig. 8g, i**). These results confirmed that NIL-IM-Lip+PD-1+L reversed the suppressive tumour-lymph node immune microenvironment (TLIME) and established antitumour immunity.

In the CT26 model, NIL-IM-Lip+PD-1+L showed stronger antitumour efficacy and better immune response. The combination therapy NIL-IM-Lip+PD-1+L group greatly inhibited the tumour growth and showed the smaller tumour volume and lighter tumour weight than the single therapy of NIL-IM-Lip+L group ($P < 0.01$, $P < 0.01$) and PD-1 group ($P < 0.001$, $P < 0.001$) (**Fig. 8k-n**). Similarly, NIL-IM-Lip+PD-1+L group showed higher frequency of CD8⁺ T cells, CTLs and NK cells compared with the NIL-IM-Lip+L group ($P < 0.01$, $P < 0.05$, $P < 0.01$) and the PD-1 group ($P < 0.01$, $P < 0.001$, $P < 0.001$) in the tumour tissues (**Fig. 8o,q**). The frequency of CD4⁺ T cells in the NIL-IM-Lip+PD-1+L group was higher than that in the PD-1 group, but no difference was found between NIL-IM-Lip+PD-1+L group and NIL-IM-Lip+L group. NIL-IM-

Lip+PD-1+L group greatly suppressed the immunosuppressive Treg cells than NIL-IM-Lip+L group ($P < 0.05$) and PD-1 group ($P < 0.001$). For the evaluation of immune cells in the lymph nodes (Fig. 8p,r), NIL-IM-Lip+PD-1+L group activated more CD4⁺ T cells, CD8⁺ T cells and NK cells and inhibited more Treg cells.

These results verified that NIL-IM-Lip combined with PD-1 mAb showed great antitumour immunotherapy in both B16F10 tumours and CT26 tumours by remodelling the suppressive TLIME and simultaneously comobilizing T and NK cells in the TLIME.

Fig. 8. Amplified antitumour effects of NIL-IM-Lip+L and PD-1 mAb cotreatment in hot tumours (B16F10 model) and cold tumours (CT26 model). Mice were sacrificed when the tumour volumes reached ~2000 mm³. **a**, Schedule of the NIL-IM-Lip+L combined with PD-1 mAb experiment in B16F10 tumour-bearing C57BL/6 mice. **b**,

Tumour volume curves of different formulations (n = 5 biologically independent animals per group). **c**, Photographs of tumours after 16 days of treatment (n = 5 biologically independent animals per group). **d**, Tumour weights in different formulation-treated groups (n = 5 biologically independent animals per group). **e**, Individual tumour volume curves after treatment with different formulations (n = 5 biologically independent animals per group). **f, h**, Flow cytometric analysis of the intratumoural CTLs, CD4⁺ T cells and CD8⁺ T cells (h), NK cells and Treg cells (n = 3 biologically independent experiments per group). **g, i**, Flow cytometric analysis of the infiltration of DCs, CD4⁺ T cells, CD8⁺ T cells (i), NK cells and Treg cells in the LNs (n = 3 biologically independent experiments per group). **j**, Schedule of the NIL-IM-Lip+L combined with PD-1 mAb antitumour experiment in CT26 tumour-bearing BALB/c mice. **k**, Tumour volume curves of different formulations (n = 5 biologically independent animals per group). **l**, Photographs of tumours after 16 days of treatment (n = 5 biologically independent animals per group). **m**, Tumour weights in different formulation-treated groups (n = 5 biologically independent animals per group). **n**, Individual tumour volume curves after treatment with different formulations (n = 5 biologically independent animals per group). **o, q**, Flow cytometric analysis of the intratumoural CTLs, CD4⁺ T cells, CD8⁺ T cells (q), NK cells and Treg cells (n = 3 biologically independent experiments per group). **p, r** Flow cytometric analysis of the infiltration of DCs, CD4⁺ T cells and CD8⁺ T cells (r), NK cells and Treg cells in the LNs (n = 3 biologically independent experiments per group). Statistical significances were calculated by one-way ANOVA analysis of variance with Tukey's post hoc test. **P* < 0.05, ***P* < 0.01, ****P* < 0.001.

MINOR:

1. In the study, there are lots of different types of formulations. Although abbreviations were given in the text, they should also be given in the figure legends to make it clearer for the readers.

Response: Thanks for your comments. The abbreviations of different types of formulations were also supplemented in the figure legends as your suggestions. The revised parts were marked in red in the revised manuscript.

REVIEWERS' COMMENTS

Reviewer #1 (Remarks to the Author):

They have addressed the previous reviews.

Reviewer #3 (Remarks to the Author):

The authors' responses and revisions are appropriate.